# CD8$^+$ regulatory T cells are critical in prevention of autoimmune-mediated diabetes

Chikako Shimokawa[1,2,3 ✉], Tamotsu Kato[3,4], Tadashi Takeuchi[3,5], Noriyasu Ohshima [6], Takao Furuki[7], Yoshiaki Ohtsu[8], Kazutomo Suzue[2], Takashi Imai[2], Seiji Obi[2], Alex Olia[1,2], Takashi Izumi [6], Minoru Sakurai[7], Hirokazu Arakawa[8], Hiroshi Ohno [3,4,9 ✉] & Hajime Hisaeda[1,2 ✉]

Type 1 diabetes (T1D) is an autoimmune disease in which insulin-producing pancreatic β-cells are destroyed. Intestinal helminths can cause asymptomatic chronic and immunosuppressive infections and suppress disease in rodent models of T1D. However, the underlying regulatory mechanisms for this protection are unclear. Here, we report that CD8$^+$ regulatory T (Treg) cells prevent the onset of streptozotocin -induced diabetes by a rodent intestinal nematode. Trehalose derived from nematodes affects the intestinal microbiota and increases the abundance of *Ruminococcus* spp., resulting in the induction of CD8$^+$ Treg cells. Furthermore, trehalose has therapeutic effects on both streptozotocin-induced diabetes and in the NOD mouse model of T1D. In addition, compared with healthy volunteers, patients with T1D have fewer CD8$^+$ Treg cells, and the abundance of intestinal *Ruminococcus* positively correlates with the number of CD8$^+$ Treg cells in humans.

[1] Department of Parasitology, National Institute of Infectious Disease, Tokyo 162-8640, Japan. [2] Department of Parasitology, Graduate School of Medicine, Gunma University, Maebashi 371-8511, Japan. [3] Laboratory for Intestinal Ecosystem, RIKEN Center for Integrative Medical Sciences, Yokohama 230-0045, Japan. [4] Immunobiolgy Laboratory, Graduate School of Medical Life Science, Yokohama City University, Yokohama 230-0045, Japan. [5] Graudate School of Medicine, Keio University, Tokyo 160-8582, Japan. [6] Department of Biochemistry, Graduate School of Medicine, Gunma University, Maebashi 371-8511, Japan. [7] Center for Biological Resources and Informatics, Tokyo Institute of Technology, Yokohama 226-8502, Japan. [8] Department of Pediatrics, Graduate School of Medicine, Gunma University, Maebashi 371-8511, Japan. [9] Intestinal Microbiota Project, Kanagawa Institute of Industrial Science and Technology, Ebina 243-0435, Japan. ✉email: chikakos@nih.go.jp; hiroshi.ohno@riken.jp; hisa@niid.go.jp

In type 1 diabetes (T1D), an autoimmune disease, insulin-producing pancreatic β-cells are destroyed, resulting in hyperglycaemia due to insulin insufficiency. Considering the recent increase of T1D in developed countries overwhelming rate of genetic changes, environmental factors appear to affect auto-immunity. One possible explanation for the involvement of environmental factors is the 'hygiene hypothesis', which suggests that reduced exposure to pathogens because of improved hygiene increases the risk of inflammatory disorders such as autoimmunity[1,2]. Among these pathogens, parasitic helminths can cause asymptomatic chronic infections and their absence is thought to be a contributor to the hygiene hypothesis[3]. Epidemiological and geographical evidence demonstrates the inverse correlation between helminthic manifestation and T1D prevalence[4,5].

Intestinal helminthic infections are immunologically unique to induce type 2 responses as well as various regulatory immune responses to suppress host immunity for their survival within the hosts[6–8]. Animal models of T1D also support the ability of intestinal helminthic infections to prevent diabetes. Infection with *Trichinella spiralis* of non-obese diabetic (NOD) mice reduces onset of spontaneous development of diabetes by inducing dominant Th2 responses[9]. NOD mice infected with *Heligmosomoides polygyrus* (Hp) develop T1D to a lesser degree, and suppressive effects are not dependent on IL-10 or CD4+ Treg cells[10]. However, IL-10 is reported to have important functions in IL-4-deficient NOD mice[11]. This nematode also suppresses streptozotocin (STZ)-induced diabetes, and the protection is independent of IL-10 or Th2 polarisation through IL-4 signalling[12]. Aside from live helminth infection, several reports demonstrate that products and/or antigens derived from blood flukes and lymphatic filariae have the ability to suppress disease a in model of T1D[13,14]. However, such products have not been found in intestinal helminthic infections. Thus, molecular and cellular regulatory mechanisms underlying protection against T1D in intestinal helminthic infections are not clear.

As another environmental factor for increased prevalence of inflammatory disorders, recent studies indicate that the intestinal microbiota is associated with onset of some diseases. Human cohort studies demonstrate association between microbiota and T1D[15], and animal models support the notion that microbiota is involved in T1D onset[16,17]. Given that intestinal helminthes affect composition of microbiota in mice[18], protective effects of intestinal helminthes may be attributed to alteration of intestinal microbiota.

Here we show that a rodent intestinal nematode can prevent the onset of STZ-induced diabetes in a CD8+ regulatory T (Treg) cell-dependent manner. Infection with the nematode and its derivative, trehalose, affects the intestinal microbiota, resulting in the induction of CD8+ Treg cells. *Ruminococcus* spp. are more abundant in infected mice and seem to be responsible for induction of CD8+ Treg cells. Trehalose has a therapeutic effect not only in STZ-treated mice, but also in NOD mice. Furthermore, compared with healthy volunteers, patients with T1D have fewer CD8+ Treg cells and intestinal *Ruminococcus*.

## Results

### Hp infection induces CD8+ Treg cells to prevent STZ-induced diabetes

Injection of C57BL/6 mice with multiple low doses of STZ resulted in hyperglycaemia and lower plasma insulin levels at 14 days after the first STZ administration (Fig. 1a, b). Immunohistochemical analyses revealed that these mice lost insulin-producing β-cells (Fig. 1c). Thus, as widely accepted[19,20], the manipulation served as a model for autoimmune-mediated T1D. Mice infected with an intestinal nematode, *Heligmosomoides polygyrus* (Hp), at 2 weeks before T1D induction showed mild elevation of blood sugar and maintained insulin concentrations consistent with conservation of β-cells (Fig. 1a–c). These results demonstrate that infection with Hp protects mice from developing STZ-induced diabetes. Hp infection induces several immune suppressive cell types such as Foxp3+CD4+ regulatory T cells (CD4+ Treg cells) that suppress T1D in various settings[21,22]. Indeed, CD4+ Treg cells were increased in the spleen of mice infected with Hp (Supplementary Fig. 1a). However, these cells were not involved in the suppression of T1D observed in Hp-infected mice, because protective effects were not abolished in Hp-infected mice depleted of CD4+ Treg cells using an anti-CD25 antibody (Supplementary Fig. 1b).

We next examined CD8+ Treg cells identified as CD8+ T cells expressing CD122 (IL-2Rβ chain)[23,24]. As a result, Hp infection increased CD8+ Treg cells significantly in the pancreatic LN and spleen (Fig. 1d–f). Depletion of CD8+ Treg cells in Hp-infected mice by treatment with an anti-CD122 antibody completely reversed the protective effects of Hp infection against T1D (Fig. 1g–i). Although the depletion was not complete (with ~20% of these cells remaining), this depletion of CD8+ Treg cells was enough to prevent the onset of diabetes. However, the CD122+CD8− population that was also depleted by the anti-CD122 antibody might play a suppressive role in T1D development (Fig. 1f). To exclude this possibility, we performed a CD8+ Treg cells transfer experiment. Mice that received CD8+ Treg cells, but not CD122−CD8+ T cells, from Hp-infected mice did not exhibit blood glucose elevation (Fig. 1j). These results indicate that CD8+ Treg cells are responsible for the suppression of T1D. In addition, aged mice with more CD8+ Treg cells confirmed the involvement of CD8+ Treg cells in T1D suppression. As reported previously[25], 60-week-old mice had substantially more CD8+ Treg cells in their spleen than young mice (Supplementary Fig. 2a). These aged mice were resistant to diabetes induction (Supplementary Fig. 2b, c), which depended on CD8+ Treg cells because aged mice depleted of CD8+ Treg cells developed diabetes comparable with young mice (Supplementary Fig. 2d).

Functionally, an in vitro T cell-suppression assay revealed that CD8+ Treg cells from Hp-infected mice remarkably suppressed the proliferation of CD4+ and CD8+ potential effector T cells in the presence of antigen-presenting cells in contrast to those from uninfected mice showing marginal suppression (Fig. 1k). In addition, CD8+ Treg cells showed a stronger ability to suppress interferon (IFN)-γ production crucial for the development of STZ-induced diabetes[26] after Hp infection (Fig. 1l), indicating that Hp augments the suppressive functions of CD8+ Treg cells. This suppression may decrease IFN-γ-producing T cells in the pancreas of Hp-infected mice after T1D induction (Supplementary Fig. 3). Because CD8+ Treg cell addition regardless of the mouse origin increased the amount of IL-10 in culture supernatants, CD8+ Treg cells appear to secrete this anti-inflammatory cytokine (Fig. 1l). Nevertheless, the contribution of IL-10 to T1D suppression was limited (Supplementary Fig. 4).

### Trehalose produced in Hp is crucial for diabetes suppression

In terms of the molecular mechanisms of CD8+ Treg cell induction, Hp-derived molecule(s) are hypothesised to modulate intestinal environments. To test this hypothesis, we comprehensively analysed intestinal contents by gas chromatography/mass spectrometry (GC/MS). Univariate analyses of 48 identified metabolites were performed, and a volcano plot demonstrated that trehalose, a disaccharide consisting of two glucose molecules, was the most remarkably increased after Hp infection (Fig. 2a). This disaccharide was the only metabolite increased significantly as assessed by Bonferroni's method (Supplementary Table 1).

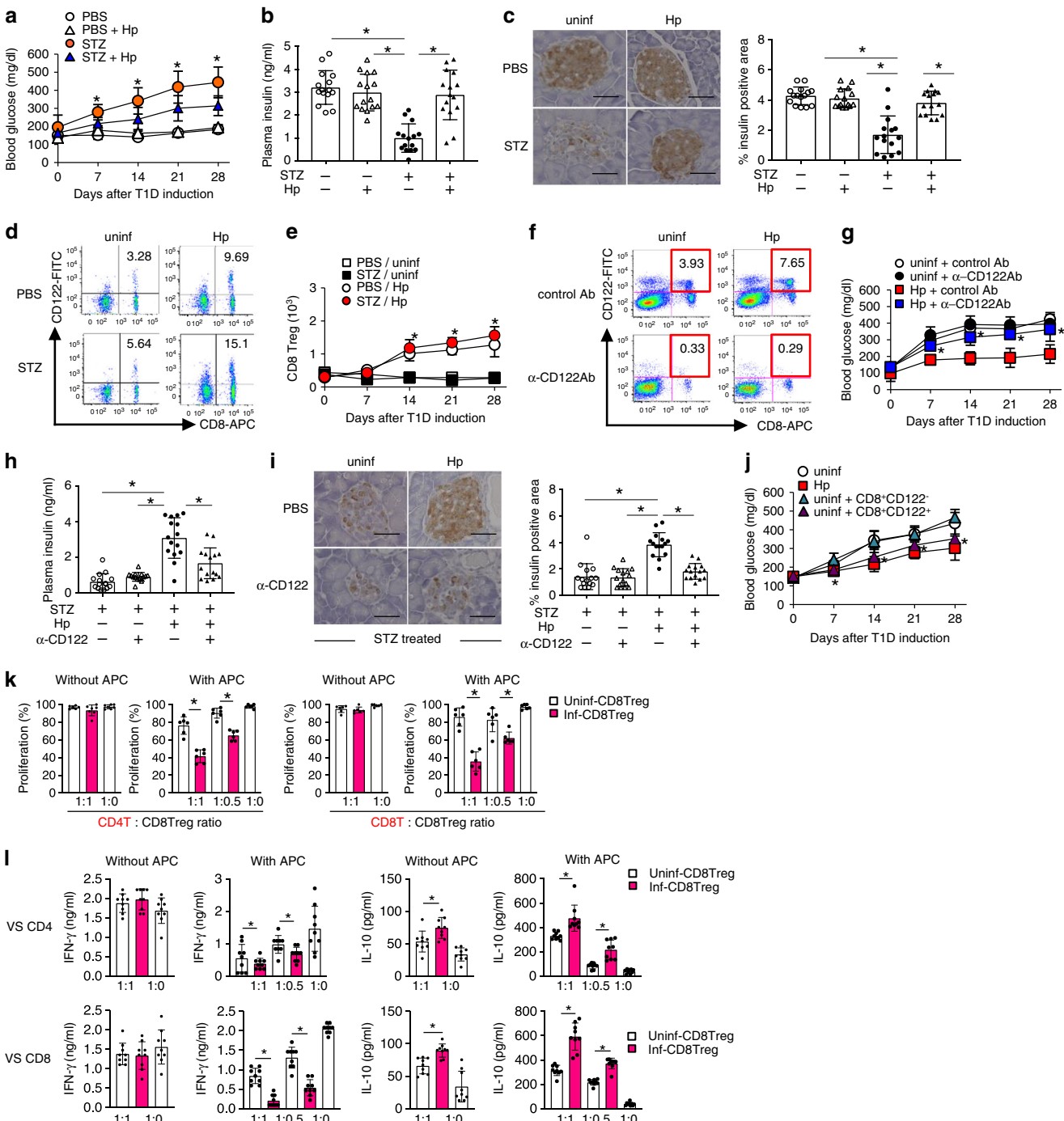

**Fig. 1 CD8$^+$ Treg cells mediate suppression of STZ-induced diabetes by *H. polygyrus*. a–c** Mice were administered STZ at 14 days after infection with Hp.
**a** Blood glucose concentrations were monitored, **b** plasma insulin was measured, and **c** pancreatic sections were stained with an anti-insulin antibody at 14 days after T1D induction. Representative histological images are shown (*left panels*), and a bar graph depicts the percentage of the stained area observed under a microscope (*right panel*). **d** CD8$^+$ Treg cells defined as CD8$^+$CD122$^+$ cells in the pancreatic LN from mice before and at 14 days after infection with Hp were quantified by flow cytometry. The *numbers* indicate the percentages of CD8$^+$ Treg cells in the FSC/SSC-gated lymphoid cells. **e** Kinetics of the absolute number of CD8$^+$ Treg cells in the pancreatic LN. **f–h** Hp-infected mice were administered an anti-CD122 antibody immediately before and after T1D induction. **f** Spleen cells of these mice were assessed for the depletive effects of the antibody on CD122-expressing cells by flow cytometry. The effects of this manipulation on blood glucose (**g**), plasma insulin levels (**h**), and pancreatic β-cells (**i**) were evaluated as described in **a–c**. **j** Blood glucose of mice that received CD8$^+$ Tregs or non-Treg CD8$^+$CD122$^-$ cells was monitored after injection of STZ. **k** TCR-driven proliferation of CD4$^+$ (*left panels*) and CD8$^+$ T (*right panels*) cells in the presence or absence of antigen-presenting cells cultured with CD8$^+$CD122$^+$ cells from the indicated mice at the indicated ratio was evaluated by flow cytometry. **l** Cytokine concentrations were quantified in supernatants of the cultured cells in **k**. Values represent the mean ± SD of 15 mice (sum of three repeated experiments, five mice each). Experiments in **l** and **k** were repeated three times, and values represent the mean ± SD of 10 mice (sum of three repeated experiments, three or four mice each). *Asterisks* denote statistical significance at *p* < 0.05 calculated by the two-way ANOVA (**a**, **e**, **g**, **j**) and Tukey post-hoc analysis (**b**, **c**, **h**, **i**, **k**, **l**). Scale bars indicate 40 μm (**c**, **i**). All experiments were repeated at least three times with similar results.

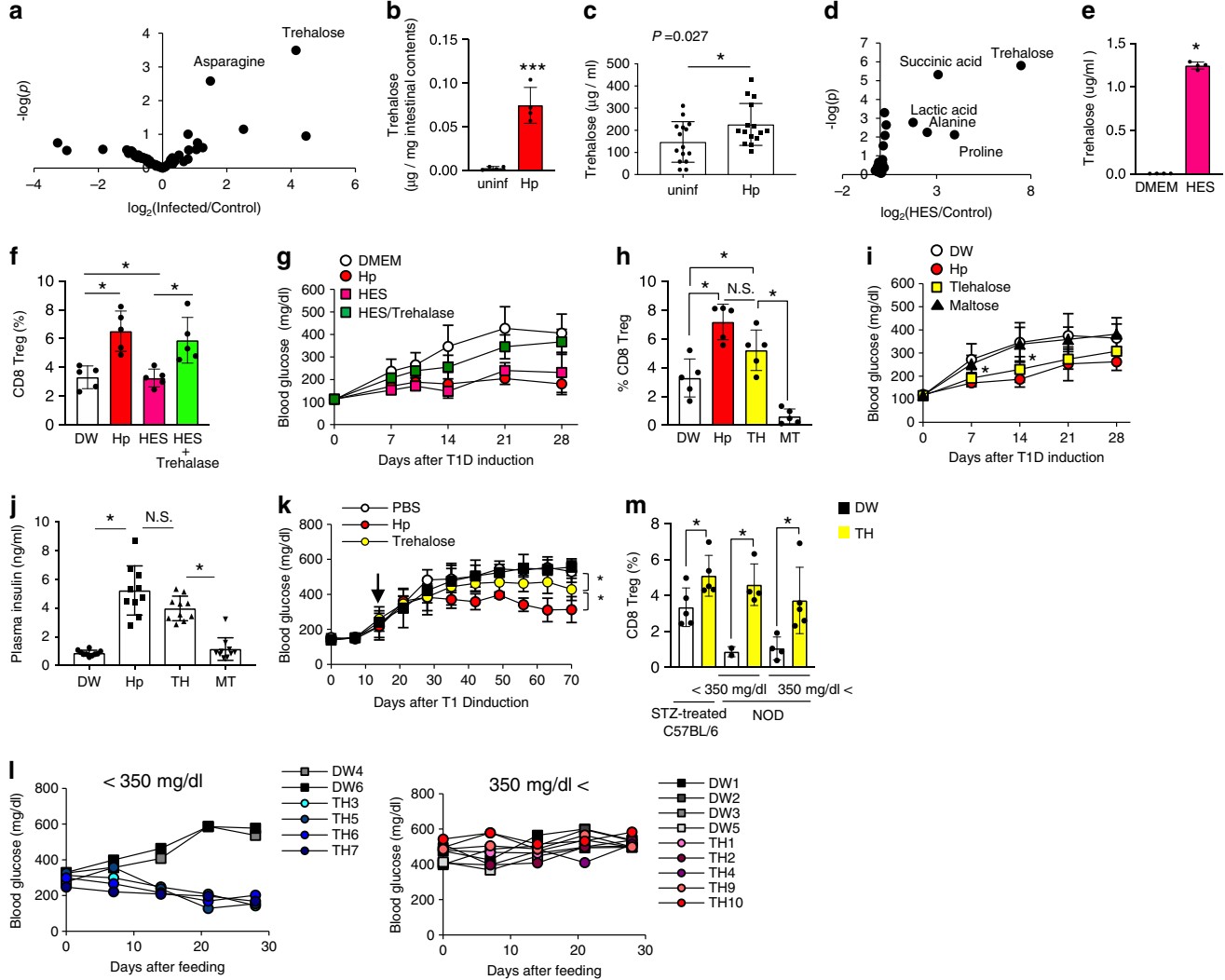

**Fig. 2 Trehalose derived from Hp induces CD8⁺ Treg cells. a** Forty-eight metabolites identified in the contents of the small intestines of five uninfected and four Hp-infected mice (**a**), and 33 metabolites in DMEM used to culture adult worms [Hp excretory/secretory (HES) antigens] (**d**) using GC/MS were subjected to univalent analyses, and volcano plots are shown. Trehalose in the small intestines of **b** and trehalose concentrations among HES antigens (**e**) were measured by GC/MS. **c** Trehalose concentrations in serum from the indicated mice were measured using ELISA. **f** Splenic CD8⁺ Treg cells in mice orally administered HES antigens or HES antigens exposed to trehalase and mice infected with Hp were analysed as described in Fig. 1d, and the frequency is shown. **g** Blood glucose concentrations were monitored in these mice after T1D induction. **h–j** Mice fed trehalose (TH) and maltose (MT) as a control disaccharide were subjected to T1D induction. Hp-infected (Hp) and uninfected (DW) mice were used as positive and negative controls, respectively. Splenic CD8⁺ Treg cells were quantified 2 weeks after feeding as described in Fig. 1d, and the percentages of these cells are shown (**h**). Blood glucose (**i**) and plasma insulin levels (**j**) were analysed as described in Fig. 1a, b. **k** Female NOD mice were fed with TH after hyperglycaemic onset. **l** Glucose levels of individual mice with hyperglycaemia of <350 mg/dl (*left panel*) or more than 350 mg/dl (*right panel*) at the beginning of sugar feeding. **m** Splenic CD8⁺ Treg cells in NOD mice used in **l** were quantified at 2 weeks after feeding trehalose as described in Fig. 1d, and the percentages of these cells are shown. Values represent the mean ± S.D. of 10 mice except for five mice in **b**. *Asterisks* denote statistical significance at $p < 0.05$ calculated by the two-tailed Mann–Whitney test (**b**, **e**), two-sided unpaired Student's *t*-test (**c**), two-way ANOVA (**g**, **i**, **k**) and Tukey post-hoc analysis (**f**, **h**, **i**, **m**). NS indicates not significant. All experiments except for GC/MS run (**a**) and NOD mice (**m**, **l**) each once, were repeated at least three times with similar results.

Absolute quantification of trehalose in the intestinal contents was also performed using GC/MS. Identification of trehalose in biological samples by GC/MS is difficult because of its similarity to both the mass spectrum and retention time of methoximated maltose, a disaccharide consisting of two glucose molecules. Thus, trehalose and maltose with methoximation were analysed in detail and differentiated clearly (Supplementary Figs. 5, 6). Finally, a substantial amount of trehalose was observed in the intestinal contents of Hp-infected mice (Fig. 2b). The trehalose concentration in the serum of mice infected with Hp was significantly higher than that in uninfected mice (Fig. 2c), suggesting that trehalose is absorbed from the intestines. Furthermore,

analysis of Hp excretory/secretory (HES) antigens collected from culture supernatants of adult worms revealed that the trehalose level was highly elevated among HES antigens (Fig. 2d, e). Three metabolites including trehalose were significantly increased among HES antigens (Supplementary Table 2), indicating that Hp produced and secreted trehalose in the intestines. In addition to adult worms, infective L3 larvae secrete trehalose. Fourier transform infra-red (FTIR) microscopic analyses revealed the location of concentrated trehalose as vesicle-like red signals along the worm body surface. Thus, a large amount of trehalose was detected in the preservative water containing L3 larvae (Supplementary Fig. 7a, b).

We next analysed whether Hp-derived molecules including trehalose contribute to diabetes suppression. Oral administration of HES antigens to mice increased CD8$^+$ Treg cells and suppressed T1D onset (Fig. 2f, g). HES antigens treated with trehalase, which degrades trehalose, did not induce CD8$^+$ Treg cells or suppress diabetes (Fig. 2f, g). Moreover, comparable with Hp infection, trehalose feeding induced CD8$^+$ Treg cells, prevented blood sugar elevation, and preserved the insulin concentration. In contrast, mice fed with control sugar maltose remained susceptible to diabetes induction (Fig. 2h–j). These results indicate that trehalose derived from Hp is an important molecule in the induction of CD8$^+$ Treg cells responsible for suppressing T1D.

To assess the therapeutic effect of trehalose, it was fed to STZ-treated mice and NOD mice after development of high blood glucose. Long-term feeding of trehalose suppressed the blood glucose elevation in STZ-treated mice significantly, but at lesser degree compared with Hp infection (Fig. 2k). Trehalose feeding to NOD mice with mild hyperglycaemia (<350 mg/dl) at the beginning of feeding completely reversed the glucose level increase (Fig. 2l). It is noteworthy that treatment with trehalose increased CD8$^+$ Treg cells even in mice refractory to treatment (Fig. 2m). These results suggest that trehalose might be used to treat T1D treatment when regeneration of pancreatic β-cells is possible.

**Intestinal microbiota contributes to diabetes suppression.** Next, to determine whether the intestinal microbiota was involved in the CD8$^+$ Treg cell induction, Hp-infected mice were orally administered an antibiotic mixture or ampicillin to perturb the microbiota prior to diabetes induction. Although these treatments did not affect Hp infection (Supplementary Fig. 8), CD8$^+$ Treg cells were not increased in mice treated with antibiotics even in the presence of Hp infection (Fig. 3a), resulting in failure to suppress STZ-induced diabetes development in these mice (Fig. 3b). Thus, the microbiota is required for the CD8$^+$ Treg cell induction crucial for diabetes suppression.

To find distinct characteristics in the microbiota inducing CD8$^+$ Treg cells, we analysed the microbiota in the small intestines and faeces of mice infected with Hp and those fed with trehalose. Mice containing more CD8$^+$ Treg cells had more genus Ruminococcus than control mice (Fig. 3d, e). We further examined the relationship between the amount of CD8$^+$ Treg cells and faecal microbiota, and found that 12 and 7 genera were positively and negatively correlated, respectively. Among these 12 genera, the most highly correlated genus was Ruminococcus, an anaerobic and Gram-positive coccus (Fig. 3e). Real-time PCR confirmed that Hp-infected and trehalose-treated mice had more Ruminococcus species than untreated mice (Fig. 3f). In addition, increased intestinal Ruminococcus was observed in aged mice and NOD mice treated with trehalose containing more CD8$^+$ Treg cells (Supplementary Fig. 9). These results strongly suggest that CD8$^+$ Treg cells induction is correlated with the abundance of Ruminococcus.

To further establish the involvement of Ruminococcus in diabetes suppression and induction of CD8$^+$ Treg cell, we attempted to isolate a single Ruminococcus species, OTU718, which was increased in mice fed with trehalose. However, it was impossible, presumably because of its requirements for strict nutrition and/or highly anaerobic conditions. Thus, we used the closest relative Ruminococcus gnavus among cultivable strains. As a control, we used Faecalibacterium prausnitzii identical to OTU58, belonging to the same family of Ruminococcaceae as OTU718 and unaffected by trehalose feeding (Fig. 3g, h). Feeding R. gnavus, but not F. prausnitzii, to STZ-treated mice significantly suppressed the blood glucose elevation (Fig. 3i). Furthermore, coculture of splenocytes from uninfected mice with supernatants from R. gnavus cultures increased CD8$^+$CD122$^+$ cells (Fig. 3j). These results indicate that these particular Ruminococcus species are, at least partially, responsible for the CD8$^+$ Treg cell induction resulting in prevention of diabetes onset.

**CD8$^+$ Treg cells and gut microbiota in patients with T1D.** We extrapolated our findings on the diabetes suppressive effect of CD8$^+$ Treg cells in mice to humans. First, we analysed CD8$^+$ Treg cells in peripheral blood obtained from children with T1D. Flow cytometric analyses demonstrated that T1D patients had fewer CD8$^+$ Treg cells, defined as CD8$^+$CD122$^+$CXCR3$^+$ cells[27], than healthy volunteers (Fig. 4a, b). By contrast, there was no difference in the CD4$^+$ Treg cell frequency (Supplementary Fig. 10). We also analysed the faecal microbiota of T1D patients and found lower ratios of the family Ruminococcaceae and genus Ruminococcus than in healthy volunteers (Fig. 4c, d). Moreover, the serum trehalose concentration in T1D patients was very low compared with that in healthy volunteers (Fig. 4e), and a highly positive correlation was found between the abundance of trehalose, CD8$^+$ Treg cells, and Ruminococcus (Fig. 4f). These results suggest that CD8$^+$ Treg cells suppress T1D development and that the gut microbiota, specifically Ruminococcus, augments CD8$^+$ Treg cells in both humans and mice.

**Discussion**

In this study, we clarified a novel mechanism underlying T1D suppression during Hp infection. Trehalose derived from Hp affects the microbiota, increasing Ruminococcus specifically, resulting in the induction of suppressive CD8$^+$ Treg cells. Because trehalose is secreted from L3 larvae, its concentration is elevated immediately after infection and maintained at high levels during infection. Trehalose has cytoprotective effects and contributes to adaptations to harmful conditions in insects such as anhydrobiosis and cryptobiosis[28,29]. L3 larvae may produce trehalose to adapt to environmental fluctuations outside of the host body where they develop. Trehalose derived from Hp, in turn, exerts anti-diabetic effects through β-cell protection, which might be explained by its ability to induce CD8$^+$ Treg cells.

Trehalose does not appear to directly induce CD8$^+$ Treg cells, but indirectly induces them through an alteration of the intestinal microbiota. Because recent reports have demonstrated that specific bacteria induce specific T-cell subsets[30–32], some bacteria may activate CD8$^+$ Treg cells. Based on our results together with utilisation of trehalose as one of the assimilable sugars by Ruminococcus[33], Ruminococcus is the most likely candidate.

Notably, our findings might be applicable to clinical situations. Compared with healthy individuals, patients with T1D have fewer CD8$^+$ Treg cells in association with a smaller number of Ruminococcus and amount of trehalose. Our insights into the suppressive mechanisms of T1D may lead to prophylactic and therapeutic applications such as using trehalose and Ruminococcus strains as a prebiotic and probiotic, respectively, as well as cell transfer of autologous CD8$^+$ Treg cells differentiated from induced pluripotent stem cells.

**Methods**

**Mice.** Male C57BL/6J mice purchased from Japan SLC Inc. and female NOD mice from CLEA Japan Inc. were maintained under specific pathogen-free conditions at 23 ± 2 °C, 55 ± 5% humidity with automated controlled 12 h dark/light cycle. Mice used for experiments at 8–10 or 60 weeks of age. All animal experiments were reviewed and approved by the Committee for Ethics on Animal Experiments at the Graduate School of Gunma University (approval number 16–041). Animal experiments were conducted in accordance with the Guidelines for Animal Experiments of the Graduate School of Gunma University, and the Low (No. 105) and Notification (No. 6) of the Japanese Government.

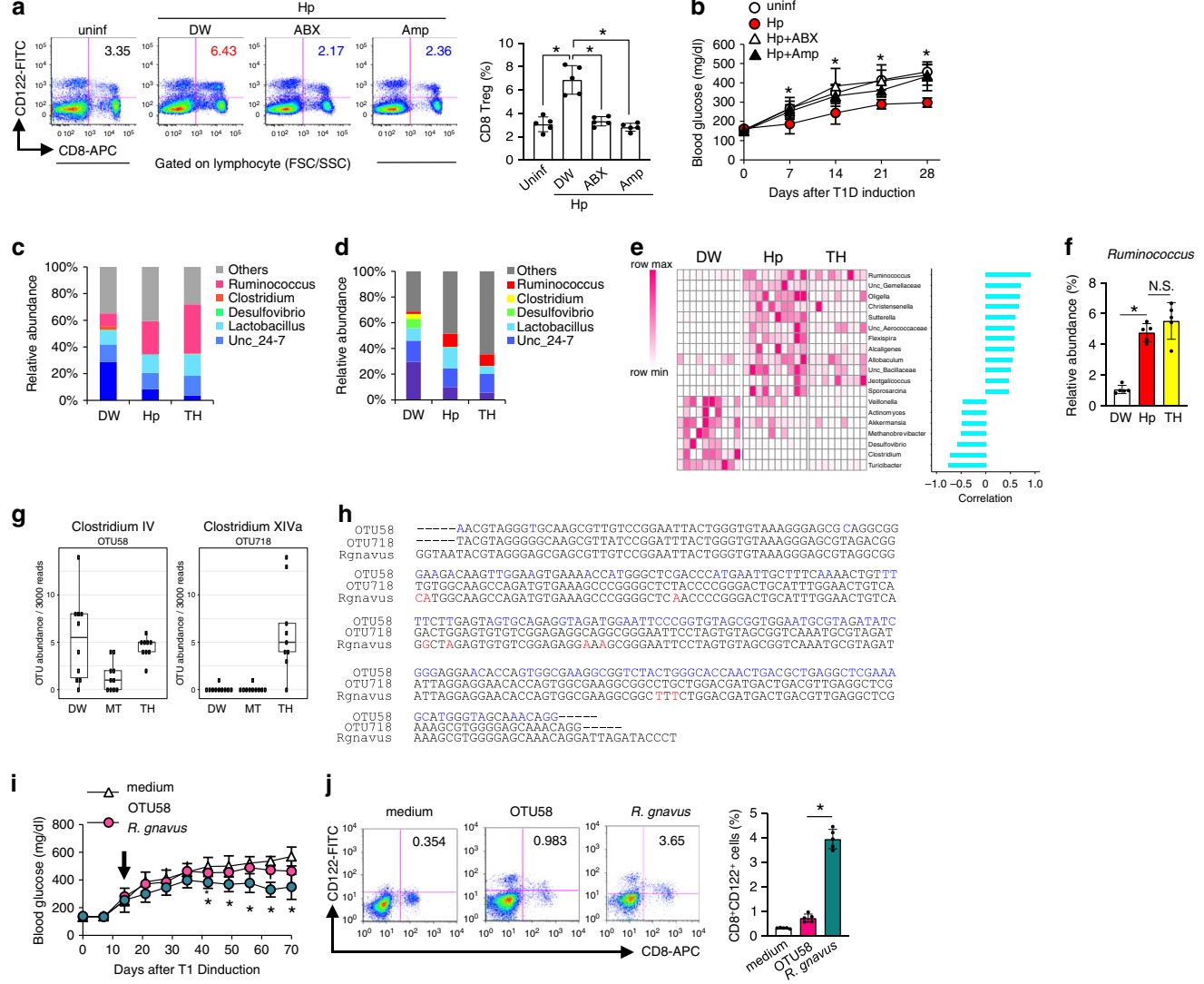

**Fig. 3 Microbiota induces of CD8+ Treg cells during Hp infection. a**, **b** Hp-infected mice treated with an antibiotic mixture (ABX), ampicillin (Amp), or untreated (DW) were used for T1D induction. CD8+ Treg cells (**a**), and blood glucose (**b**) were analysed as in Fig. 1d and a, respectively. Values represent the mean ± SD of five mice. The microbiota composition at genus levels of the small intestines (**c**) and faeces (**d**) of indicated mice at 14 days after infection or feeding. Values represent the mean of 10 (DW, Hp) or 9 (TH) mice. **e** Heatmap showing the abundance of genera of faecal bacteria correlated with the frequency of CD8+ Treg cells in mice used in **d** as depicted in the colour scale (*left panel*). Each column represents an individual animal. The positive correlation is strongest from the top (*Ruminococcus*) to the 12th row (*Sporosarcina*), and the negative correlation is strongest from the bottom (*Turicibacter*) up to the 13th row (*Veillonella*) (*right panel*). **f** Frequency of *Ruminococcus* among whole intestinal bacteria in the indicated mice re-evaluated by quantitative PCR. **g** Abundance of OTU (operational taxonomy unit) 58 and OTU718 in mice fed with TH were measured. Values represent the mean ± SD of five mice. **h** Partial DNA sequences of 18S rRNA of *Ruminococcus gnavus*, OTU58, and OTU718. Eight different nucleotides out of 257 between *R. gnavus* and OTU718 are depicted in red, and those between OTU58 and OTU718 (119/254) are depicted in blue. **i** Glucose levels were monitored in STZ-treated mice orally inoculated with OTU58 or *R. gnavus*. Values represent the mean ± SD of five mice. **j** Frequencies of CD8+ Treg cells among spleen cells cultured in the presence of culture supernatant from OTU58 or *R. gnavus* for 48 h were analysed by flow cytometry. *Numbers* in pseudocolor plots indicate the percentages of CD8+ Treg cells summarised as a bar graph. Values represent the mean ± SD of five mice. *Asterisks* denote statistical significance at $p < 0.05$ calculated by Tukey post-hoc analysis (**a**, **f**, **j**), two-way ANOVA (**b**, **i**). All experiments were repeated at least three times with similar results.

**Heligmosomoides polygyrus infection.** Hp were maintained in mice and serially passaged. For experimental infections, we used infectious L3 larvae obtained from eggs in the faeces of infected mice after culture on filter paper soaked in distilled water[34]. Mice were orally infected with 200 L3 larvae in 500 μl DW by gastric intubation. Establishment of infection was confirmed by detecting eggs in faeces.

**Induction and evaluation of diabetes.** C57BL/6J mice were intraperitoneally administered STZ (50 mg/kg body weight) for five consecutive days to induce diabetes, as described previously[12]. Blood samples were periodically collected from mice via puncture of the tail vein to monitor blood glucose concentrations using lab glucose cartridge and sensor devices (ForaCare Inc.). The determination of insulin levels in serum samples was performed by an LBIS mouse Insulin ELISA kit (AKRIN-011RU, Shibayagi Co. Ltd.), according to the manufacturer's instructions.

**Immunohistochemical examinations.** Pancreatic tissues excised from mice after STZ administration were fixed in 4% paraformaldehyde and embedded in paraffin. Tissue sections (5-μm thick) were subjected to immunohistochemistry with a polyclonal guinea pig anti-insulin antibody (A0564, Dako) at 1:200 dilution. Stained areas were quantified using a BZ-8100 microscope (Keyence), NIS-Elements (Nikon), and ImageJ (NIH)[35]. At least 10 sections from individual mice were examined.

**Flow cytometry.** Single-cell suspensions of mouse spleens, mesenteric lymph nodes, pancreatic lymph nodes, and pancreatic tissues were incubated with an anti-CD16/32 (93; eBioscience) to block Fc receptors to prevent non-specific antibody binding and then stained with the following mAbs conjugated to fluorescein iso-thiocyanate (FITC), phycoerythrin (PE), allophycocyanin (APC), phycoerythrin-

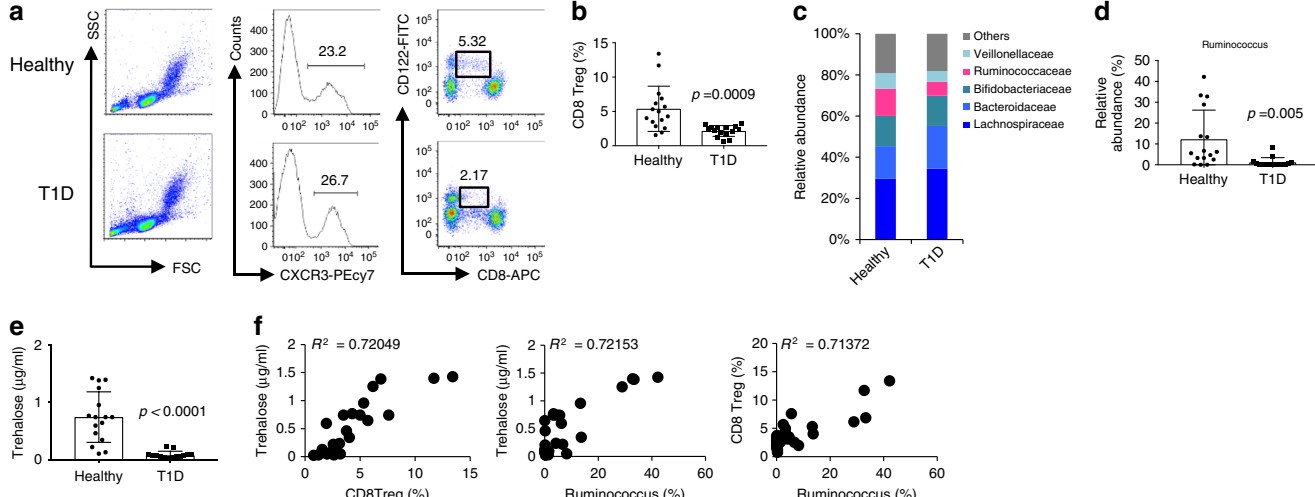

**Fig. 4 Patients with T1D have fewer CD8⁺ Treg cells compared with healthy volunteers.** Evaluation of CD8⁺ Treg cells and microbiota in T1D patients ($N = 15$) and healthy volunteers ($N = 16$) was performed. **a** Peripheral blood mononuclear cells obtained from T1D patients were stained with fluorescent dye-labelled anti-CD8, anti-CXCR3, and anti-CD122 antibodies. CXCR3⁺ cells among gated lymphoid cells (*left and centre panels*) were separated into CD8⁺ and CD122⁺ (*right panels*). The numbers indicate the percentages of gated cells. **b** Frequency of CD8⁺ Treg cells defined as CXCR3⁺CD8dullCD122⁺ cells in T1D patients and healthy volunteers is plotted as a scatter graph with bars. **c** Composition of the intestinal microbiota in T1D patients and healthy volunteers at the family level. **d** Frequency of genus *Ruminococcus* in whole intestinal bacteria. **e** Trehalose concentration in serum from T1D patients and healthy volunteers. Values represent the mean ± SD. **f** Representative co-plotted frequency of CD8⁺ Treg cells, abundance of *Ruminococcus*, and trehalose concentration in T1D patients and healthy volunteers. $R^2$ denotes the correlation coefficient. *p*-values were calculated using the two-tailed Mann–Whitney test (**b**, **d**, **e**). All experiments using human samples were performed once.

indotricarbocyanine (PE-Cy7), allophycocyanin-indotricarbocyanine (APC-Cy7), or PerCP-cy5/5 (eBioscience or BioLegend): anti-mouse CD4 (GK1.5), anti-mouse CD25 (PC61), anti-mouse CD8 (53-6.7), anti-mouse CD122 (TMβ-1), and anti-mouse IFN-γ (XMG1.2). Mononuclear cells separated from peripheral blood of T1D patients by gradient centrifugation using Ficoll-Hypaque (GE healthcare, Tokyo, Japan) were stained with fluorescent dye-conjugated anti-human CD4 (RPA-T4), anti-human CD25 (BC96), anti-human CD8 (SK1), anti-human CD122 (TU27), and anti-human CXCR3 (G025H7) antibodies. For intracellular staining, cells stained as described above were fixed and permeabilized with BD Cytofix/Perm (BD Bioscience) and then stained with anti-mouse Foxp3 (MF-14) or anti-human Foxp3 (259D) antibodies. All fluorescent antibodies were used at dilution 1/50. Stained cells were collected on FACSverse (BD Bioscience) and data acquired using FACSDiva (BD Bioscience). Data analysis was performed using FlowJo 9.1 software (Treestar). Gating strategies are shown in Supplementary Fig. 11.

**In vivo cell depletion and cytokine neutralisation**. To deplete cells expressing CD122 including CD8⁺ Treg cells in vivo, mice were injected with 500 μg anti-CD122 mAb (TMβ-1) or control isotype rat IgG at 1 and 3 days before and at 7 and 14 days after the first STZ administration. CD4⁺ Treg cells were depleted using the anti-CD25 (7D4) antibody as described above. For IL-10 neutralisation, mice were injected intraperitoneally with 500 μg anti-IL-10 monoclonal antibody (JES5-2A5) at −1, 0, 5, and 7 days after T1D induction.

**Isolation and adoptive transfer of CD8⁺ Treg cells**. Single-cell suspensions of spleens from Hp-infected mice were stained with fluorescent dye-conjugated anti-CD8 and anti-CD122 antibodies. CD8⁺CD122⁺ and CD8⁺CD122⁻ cells were sorted by a FACSAria II (BD Bioscience). The sorted cells were at least 98% pure. Five hundred thousand purified cells were intravenously transferred into uninfected recipient mice at 1 day before STZ administration.

**In vitro T cell-suppression assay**. Briefly, purified splenic CD4⁺CD25⁻ or CD8⁺CD122⁻ responder cells from uninfected mice were labelled using a CellTrace Violet kit (Thermo Fisher). The cells were then cocultured with sorted CD8⁺ Treg cells from uninfected or Hp-infected mice with or without antigen-presenting cells (splenic CD3⁻CD8⁻ cells) from uninfected mice in the presence of a plate-bound anti-CD3 antibody (2C11) for 3 days. Cells were harvested and analysed by flow cytometry. Labelled cells with diluted fluorescence were considered as proliferative cells. Cytokines in supernatants of cell cultures were also analysed using ELISA kits (R&D Systems), according to the manufacturer's instructions.

**GC–MS analysis**. Contents in the small intestines of mice were collected in Eppendorf tubes on ice and then weighed. Then, 250 μl of a solvent mixture (MeOH:H₂O:CHCl₃ = 2.5:1:1) and 5 μl of 1 mg/ml 2-isopropylmalic acid (2-IPM)

(Sigma-Aldrich) as an internal standard were added to the tube. The mixture was vortexed for 30 min at room temperature before centrifugation at 21,000×*g* for 5 min at room temperature. The supernatant (225 μl) was transferred to a new tube, and 200 μl of water was added to the tube. After vortexing, the tube was centrifuged at 21,000×*g* for 5 min at room temperature, and 250 μl of the supernatant was transferred to a new tube and stored in a freezer before use. The supernatant (50 μl) was transferred to a new tube and lyophilised using a centrifugal concentrator. For oximation, 40 μl pyridine with or without 20 mg/ml methoxyamine hydrochloride (Sigma-Aldrich) was added to the lyophilised sample. The tube was sonicated to disperse the lyophilised powder before shaking at 1400 rpm for 90 min at 30°C. Then, 20 μl *N*-methyl-*N*-trimethylsilyl-trifluoroacetamide (MSTFA) (GL Science) was added for derivatization. The mixture was then incubated at 37 °C for 30 min with shaking at 1400 rpm. The tube was centrifuged at 21,000×*g* for 5 min at room temperature, and 1 μl of the resultant supernatant was injected into a DB-5 capillary column (30 × 0.25 mm; film thickness: 1 μm) (Agilent Technologies). In addition, GC/MS analysis was performed using a GCMS-TQ8030 (Shimadzu) equipped with an AOC-20i autosampler (Shimadzu).

Analysis of small molecular weight metabolites was performed based on Smart Metabolites Database Release 3.01 (Shimadzu) that contains the data acquisition parameters for 571 compounds in full-scan mode and 467 compounds in multiple reaction monitoring (MRM) mode. Data acquisition was performed in both full-scan and MRM modes. GC–MS solution software Version 4.41 (Shimadzu) was used for data processing. Retention time correction was performed based on the retention time of a standard *n*-alkane mixture (Restek). The peaks were assigned automatically and checked manually. For comparison between samples from control and infected mice, each peak area was normalised based on the weight of intestinal contents and the peak area of 2-IPM. Statistical analysis was performed using the two-tailed unpaired Student's *t*-test. *p*-values were adjusted by Bonferroni's method and the Benjamini–Hochberg method.

**Measurement of trehalose**. Trehalose measurement was performed in L3 larval samples and human sera using a trehalose assay kit (#K-TREH, Magazyme), according to the manufacturer's instructions.

**Preparation of HES antigens**. Adult worms collected from the small intestines of Hp-infected mice were washed extensively in sterile PBS containing penicillin and streptomycin (Gibco), and 200 worms were cultured in 1 ml DMEM (Sigma-Aldrich) containing penicillin and streptomycin for 3 days. The supernatant was collected as HES antigens. In some experiments, trehalase (Sigma-Aldrich) was added to HES antigens at 0.025 U/ml, followed by incubation overnight at 37 °C[36].

**Antibiotic treatments**. For antibiotic treatments, mice were treated with the following combination of antibiotics (ABX): ampicillin (1 g/l), metronidazole (1 g/l),

vancomycin (500 mg/l), and neomycin (1 g/l), or ampicillin (1 g/l) alone (Amp) in drinking water for 14 days.

**Trehalose feeding**. Mice were fed 3% trehalose in drinking water for 7 days before STZ treatment or 500 µl HES antigens with or without trehalose exposure by gastric intubation for 7 days.

**FTIR measurements**. The FTIR measurements of infected L3 larvae of Hp were performed according to a previous study[37]. The whole body of larvae was sandwiched between two KBr plates. Lattice mapping spectra in the 4000–750 cm⁻¹ range were collected by an infra-red microscope (IRT-7200 with FT/IR-6600 spectrometer; JASCO) equipped with a liquid nitro-gen-cooled, mercury-cadmium-telluride, 16-element, linear array detector. Sequential spectra were collected at 570 points (15 × 38 points) in the specimen. For each spectrum, 32 interferograms were collected, signal averaged, and Fourier transformed to generate spectra with a spectral resolution of 8 cm⁻¹, pixel reso-lution of 12.5, and pixel resolution of 12.5 signal averaged, and Fourier transformed to generate spectra with a spectral resolution of 8 interferograms.

**16S rRNA gene pyrosequencing**. Faecal and small intestinal samples collected from mice were immediately frozen in liquid nitrogen and stored at −80 °C. Faecal DNA extraction was performed according to a previous study[38] with minor modifications. A grain of mouse faeces or human faecal pellets were suspended with sterilised sticks in 475 µl TE10 buffer containing 10 mM Tris-HCl (pH 8.0) and 10 mM EDTA. The faecal suspension was incubated with 15 mg/ml lysozyme (Wako) at 37 °C for 1 h. A final concentration of 2000 U/ml purified achromo-peptidase (Wako) was then added, followed by incubation at 37 °C for 30 min. Then, 1% (wt/vol) sodium dodecyl sulfate and 1 mg/ml proteinase K (Merck Japan) were added to the suspension, followed by incubation at 55 °C for 1 h. After centrifugation, bacterial DNA was purified using a phenol/chloroform/isoamyl alcohol (25:24:1) solution. The DNA was precipitated by adding ethanol and sodium acetate. RNase A (Wako) was added to bacterial DNA in TE buffer to a final concentration 1 mg/ml. To remove fragmented low molecular weight DNA, polyethylene glycol (PEG 6000) precipitation was performed after RNase treatment.

The V4 variable region (515F–806R) was sequenced on an Illumina MiSeq, following the method of Kozich et al.[39] Each reaction mixture contained 15 pmol of each primer, 0.2 mM deoxyribonucleoside triphosphates, 5 µl of 10× Ex Taq HS buffer, 1.25 U Ex Taq HS polymerase (Takara), 50 ng extracted DNA, and sterilised water to reach a final volume of 50 µl. PCR conditions are as follows: 95 °C for 2 min, 25 cycles of 95 °C for 20 s, 55 °C for 15 s, and 72 °C for 1 min, followed by 72 °C for 3 min. The PCR product was purified by AMPure XP (Beckman Coulter) and quantified using a Quant-iT PicoGreen ds DNA Assay Kit (Life Technologies Japan). Mixed samples were prepared by pooling approximately equal amounts of PCR amplicons from each sample. The pooled library was analysed with an Agilent High Sensitivity DNA Kit on an Agilent 2100 Bioanalyzer (Agilent Technologies). Real-time PCR for quantification was performed on the pooled library using a KAPA Library Quantification Kit for Illumina, following the manufacturer's protocols. Based on the quantification, the sample library was denatured and diluted. A sample library with 20% denatured PhiX spike-in was sequenced by MiSeq using a 500-cycle kit. We obtained 2 × 250 bp paired-end reads. The sequence data were processed using Quantitative Insights into Microbial Ecology software (QIIME, v1.8.0) and Mothur v. 1.36.1[40].

**Real-time quantitative PCR**. Bacterial genomic DNA was isolated from faecal pellets using a QIAamp Stool Mini Kit (Qiagen). DNA encoding 16S rRNA was quantified by SYBR Green dye incorporation (Takara) analysed using an ABI Prism 7700 thermal cycler and detector system (Thermo Fisher Scientific)[41]. qPCR was carried out according to the manufacturers' instructions. The PCR primer sequences used to universally amplify 16S rRNA of all bacteria were 5′-GTGCCAGCMGCCGCGGTAA-3′ and 5′-GACTACCAGGGTATCTAAT-3′. The sequences used to specifically amplify 16S rRNA of *Ruminococcus* were 5′-CTAGGTGAAGATACTGACGGTAACCTG-3′ and 5′-GTAT-TACCGCGGCTGCTGGCAC-3′[42]. The relative amount of *Ruminococcus* to whole bacteria was calculated based on the difference in the threshold cycle between universal and specific PCR products.

**Bacterial culture**. *Ruminococcus gnavus* (JCM6515), the closest species to OTU718, and *Faecalibacterium prausnitzii* (JCM 31915) identical to OTU58 were obtained from the RIKEN BioResource Research Center. Both bacteria were cul-tivated in YCFA medium[43]. The media were centrifuged and separated into pre-cipitates and supernatants. To adjust the concentration, the precipitates were diluted with PBS, resulting in an OD 600 of approximately 0.8 (4 × 10⁸ CFU). Supernatants were passed through membrane filters with a 0.2-µm pore size (Sartorius) and diluted to adjust the concentration in accordance with the OD 600 of precipitates before use.

**Colonisation of bacteria and bacterial stimuli of T cells**. *R. gnavus* and control bacteria *F. prausnitzii* were grown overnight, and then ~1 × 10⁸ CFU in 200 µl YCFA medium was orally administered to B6 mice at 14 days after diabetes induction for 5 days. Blood glucose levels in the mice were analysed each week. For in vitro experiments, splenocytes (1 × 10⁵) from uninfected mice were incubated with supernatants from the bacterial cultures at a medium:supernatant ratio of 4:1. All cultures were performed in triplicate wells containing 200 µl complete RPMI medium (RPMI 1640 containing 2 mM L-glutamine and 25 mM HEPES) supple-mented with 10% FBS for 2 days.

**Human samples**. The Ethics Committee of the Graduate School of Medicine, Gunma University approved all human experiments conducted in this study (approval number 2016-071). Nineteen patients and 16 healthy volunteers were enroled. Informed consent was obtained from the parents of participating children and/or participants. The clinical characteristics of the patients are summarised in Supplementary Table 3. Blood samples from newly diagnosed patients were col-lected at the inpatient department, and samples from well-controlled patients were collected at the outpatient department. All faecal samples were collected in tubes containing RNAlater (Sigma-Aldrich) within 3 days before or after blood collection and stored at 4 °C until analysis.

**Statistical analysis**. All statistical analyses were performed using Prism software with the two-tailed unpaired Student's t-test or one-way ANOVA, followed by Tukey's post-hoc test or two-tailed Mann–Whitney test. p-values of <0.05 were considered as significant (*p < 0.05, **p < 0.01, and ***p < 0.001).

## Data availability
Sequence data are available at DDBJ with the accession code PRJDB9558. The authors declare that the other data underlying the figures and Supplementary Information in this manuscript are available from the authors on reasonable request.

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

# ARTICLE

15. Giongo, A. et al. Toward defining the autoimmune microbiome for type 1 diabetes. *ISME J.* **5**, 82–91 (2011).
16. Wen, L. et al. Innate immunity and intestinal microbiota in the development of Type 1 diabetes. *Nature* **455**, 1109–1113 (2008).
17. Markle, J. G. M. et al. Gammadelta T cells are essential effectors of type 1 diabetes in the nonobese diabetic mouse model. *J. Immunol.* **190**, 5392–5401 (2013).
18. Walker, A. W. et al. Dominant and diet-responsive groups of bacteria within the human colonic microbiota. *ISME J.* **5**, 220–230 (2011).
19. Muller, A., Schott-Ohly, P., Dohle, C. & Gleichmann, H. Differential regulation of Th1-type and Th2-type cytokine profiles in pancreatic islets of C57BL/6 and BALB/c mice by multiple low doses of streptozotocin. *Immunobiol.* **205**, 35–50 (2002).
20. Paik, S. G., Blue, M. L., Fleischer, N. & Shin, S. Diabetes susceptibility of BALB/cBOM mice treated with streptozotocin. Inhibition by lethal irradiation and restoration by splenic lymphocytes. *Diabetes* **31**, 808–815 (1982).
21. Finney, C. A. M., Taylor, M. D., Wilson, M. S. & Maizels, R. M. Expansion and activation of CD4(+)CD25(+) regulatory T cells in *Heligmosomoides polygyrus* infection. *Eur. J. Immunol.* **37**, 1874–1886 (2007).
22. Aravindhan, V. et al. Decreased prevalence of lymphatic filariasis among subjects with type-1 diabetes. *Am. J. Trop. Med. Hyg.* **83**, 1336–1339 (2010).
23. Akane, K., Kojima, S., Mak, T. W., Shiku, H. & Suzuki, H. CD8+CD122+CD49d^low regulatory T cells maintain T-cell homeostasis by killing activated T cells via Fas/FasL-mediated cytotoxicity. *Proc. Natl Acad. Sci. USA* **113**, 2460–2465 (2016).
24. Endharti, A. T. et al. Cutting edge: CD8+CD122+ regulatory T cells produce IL-10 to suppress IFN-gamma production and proliferation of CD8+ T cells. *J. Immunol.* **175**, 7093–7097 (2005).
25. Rifa'i, M., Kawamoto, Y., Nakashima, I. & Suzuki, H. Essential roles of CD8+CD122+ regulatory T cells in the maintenance of T cell homeostasis. *J. Exp. Med.* **200**, 1123–1134 (2004).
26. Herold, K. C. et al. Regulation of cytokine production during development of autoimmune diabetes induced with multiple low doses of streptozotocin. *J. Immunol.* **156**, 3521–3527 (1996).
27. Shi, Z. et al. Human CD8+CXCR3+ T cells have the same function as murine CD8+CD122+ Treg. *Eur. J. Immunol.* **39**, 2106–2119 (2009).
28. Erkut, C., Gade, V. R., Laxman, S. & Kurzchalia, T. V. The glyoxylate shunt is essential for desiccation tolerance in *C. elegans* and budding yeast. *eLife* **5**, e13614 (2016).
29. Watanabe, M., Kikawada, T. & Okuda, T. Increase of internal ion concentration triggers trehalose synthesis associated with cryptobiosis in larvae of *Polypedilum vanderplanki*. *J. Exp. Biol.* **206**, 2281–2286 (2003).
30. Furusawa, Y. et al. Commensal microbe-derived butyrate induces the differentiation of colonic regulatory T cells. *Nature* **504**, 446–450 (2013).
31. Atarashi, K. et al. Treg induction by a rationally selected mixture of *Clostridia* strains from the human microbiota. *Nature* **500**, 232–236 (2013).
32. Obata, Y. et al. The epigenetic regulator Uhrf1 facilitates the proliferation and maturation of colonic regulatory T cells. *Nat. Immunol.* **15**, 571–579 (2014).
33. Crost, E. H. et al. Utilisation of mucin glycans by the human gut symbiont *Ruminococcus gnavus* is strain-dependent. *PLoS ONE* **8**, e76341 (2013).
34. Shimokawa, C. et al. Mast cells are crucial for induction of group 2 innate lymphoid cells and clearance of helminth infections. *Immunity* **46**, 863–874. e864 (2017).
35. Kikuchi, O. et al. FoxO1 gain of function in the pancreas causes glucose intolerance, polycystic pancreas, and islet hypervascularization. *PLoS ONE* **7**, e32249 (2012).
36. Johnston, C. J. C. et al. Cultivation of *Heligmosomoides polygyrus*: an immunomodulatory nematode parasite and its secreted products. *J. Vis. Exp.* **6**, e52412 (2015).
37. Sakurai, M. et al. Vitrification is essential for anhydrobiosis in an African chironomid, *Polypedilum vanderplanki*. *Proc. Natl Acad. Sci. USA* **105**, 5093–5098 (2008).
38. Atarashi, K. et al. Th17 Cell induction by adhesion of microbes to intestinal epithelial cells. *Cell* **163**, 367–380 (2015).
39. Kozich, J. J., Westcott, S. L., Baxter, N. T., Highlander, S. K. & Schloss, P. D. Development of a dual-index sequencing strategy and curation pipeline for analyzing amplicon sequence data on the MiSeq Illumina sequencing platform. *Appl. Environ. Microbiol.* **79**, 5112–5120 (2013).
40. Myer, P. R., Kim, M., Freetly, H. C. & Smith, T. P. Evaluation of 16S rRNA amplicon sequencing using two next-generation sequencing technologies for phylogenetic analysis of the rumen bacterial community in steers. *J. Microbiol. Methods* **127**, 132–140 (2016).
41. Wang, I. K. et al. Real-time PCR analysis of the intestinal microbiotas in peritoneal dialysis patients. *Appl. Environ. Microbiol.* **78**, 1107–1112 (2012).
42. Fuhrer, A. et al. Milk sialyllactose influences colitis in mice through selective intestinal bacterial colonization. *J. Exp. Med* **207**, 2843–2854 (2010).
43. Browne, H. P. et al. Culturing of 'unculturable' human microbiota reveals nove taxa and extensive sporulation. *Nature* **533**, 543–546 (2016).

## Acknowledgements

We thank Ms. Wakana Mizutani for technical assistance, Dr. Osamu Kikuchi (Metabolic Signal Research Center, Institute of Molecular and Cellular Regulation, Gunma University) for preparing pancreatic sections, and Mr Ken-ichi Akao and Taro Takami (JASCO Corporation) for assistance with FTIR imaging. We are sincerely grateful to all of the T1D patients and healthy volunteers who participated in this study. We also thank Mitchell Arico from Edanz Group (www.edanzediting.com/ac) for editing a draft of this manuscript. This work was supported by a Grant-in-Aid for International Scientific Research (B) from the Japan Society for the Promotion of Science (15H05274 to H.H.), Grants-in-Aid for Scientific Research (B) (16H05207 to H.O.) and (C) (15K08441 and JP19K07530 to H.H.), and Early career scientists (19K16682 to C.S.) from the Ministry of Education, Culture, Sports, Science, and Technology, the Japan Agency for Medical Research and Development (JP19fk018096 to H.H.), The Food Science Institute Foundation to H.O., Core Research for Evolutional Science and Technology (JP18gm0710009 to H.O.), Grants provided by the Ichiro Kanehara Foundation Japan, Takeda Science Foundation, Naito foundation, Yakult Bio-Science Foundation, Shiseido Female Researcher Science Grant, The Nakajima Foundation, and Uehara Memorial Foundation to C.S.

## Author contributions

C.S. and H.H. conceived the study. C.S. designed and performed experiments, analysed experimental data, and wrote the manuscript. T.K., T.T., and H.O. contributed to microbiotic analyses. N.O. and T.Izumi biochemically analysed intestinal contents. T.F. and M.S. performed FTIR imaging. Y.O. and H.A. recruited children with T1D. K.S., T. Imai, O.A., and S.O. organised experimental animals and helped to perform experiments. C.S., H.O., and H.H. supervised the research and wrote the manuscript.

## Competing interests

The authors declare no competing interests.
