## [Peer Review File · Nature Communications]

Reviewers' comments:

Reviewer #1 (Remarks to the Author):

This is an interesting study reporting that a rodent intestinal nematode (*Heligmosomoides polygyrus*) prevents the presentation of streptozotocin-induced diabetes through induction of CD8+ regulatory T cells (CD8Treg). The authors provide evidence that the mechanism is mediated by trehalose, a disaccharide produced by the nematode. Trehalose modulates the intestinal microbiota increasing the abundance of *Ruminococcus*, which induces CD8Treg cells. Children with type 1 diabetes (T1D) had lower trehalose concentration in their serum and reduced proportion of CD8Treg cells when compared to healthy controls.

General comment: The caveat of this paper is that streptozotocin-induced diabetes in rodents and autoimmune diabetes in NOD mice are not identical to human type 1 diabetes (T1D). Therefore I suggest that the title should be revised to read "CD8+ regulatory T cells play a critical role in prevention of streptozotocin -induced diabetes".

Specific comments:

1. The methodology used seems appropriate.
2. Do the authors have any information on the correlation between intestinal trehalose and trehalose in the peripheral circulation,
3. When looking at the supplementary table with data on the children with T1D analyzed, one can see that the disease duration is highly variable (from diagnosis to a disease duration of almost 15 years). Only two children have a disease duration of less than one year. It would have been preferable to include only children with newly diagnosed T1D given that the abundance of CD8Treg cells is hardly stable over time.
4. Ten of the 15 children with T1D have a CD8Treg count of $\geq 2\%$, which seems to be within the normal range based on Figure 4b.
5. Only GAD antibodies were analyzed in the children with T1D. The cutoff for GAD antibody positivity should be given and IA-2 and ZnT8 antibodies should as well be analyzed in these children
5. From where is the trehalose in the human serum samples derived?
6. The proportional difference in the abundance of *Ruminococcus* in the gut microbiota between children with T1D and healthy children is conspicuously smaller than the difference in serum trehalose concentrations between the two groups. Why is that?

Reviewer #2 (Remarks to the Author):

The finding that helminthic infection can protect from type 1 diabetes in mouse models is not novel, however, the novelties of this manuscript are as follows: 1) the authors identified that CD122+ CD8+ T cells induced by a mouse nematode (*Heligmosomoides polygyrus*) infection contribute to the protection from diabetes; 2) the authors identified that trehalose, possibly produced by gut commensal bacteria *Ruminococcus*, contributes to CD8 Treg induction and diabetes protection and 3) the authors also found that children with T1D have lower levels of CD122+ CD8 Treg and lower abundance of *Ruminococcus* in the fecal samples. However, this reviewer has the following concerns and comments:

- 1) This is not an ideal model: the author used a chemical, STZ, to generate an induced diabetes model, which is not an autoimmune diabetes model. The best mouse model of type 1 diabetes is the non-obese diabetic (NOD) mouse model, and although the STZ model system has been used by many investigators, the cause of human T1D is not by a beta cell toxic chemical agent. Thus additional cautions apply in the translation of the finding to humans.
- 2) In one experiment, the authors used diabetic NOD mice and found that trehalose feeding could

reverse the hyperglycemia for up to a month in about 50% of diabetic NOD mice. It is not clear if the mice that became normoglycemic expressed more CD122+ CD8 Tregs; and it is also not clear if there were changes in the composition of gut microbiota of these mice.

3) STZ needs to be dissolved in a citrate acid buffer, pH 4.0, and studies have suggested that pH, especially low pH can alter the composition of gut microbiota. The authors should use citrate buffer, not PBS (pH ~7.4), as the control, especially as the authors studied the gut microbiota in the mice.

4) CD122 is expressed on other types of immune cells including NK cells; it is not clear what the effect of anti-CD122 antibody depletion in vivo was on other types of immune cells, in addition to the CD122+ CD8 T cells. It is also not clear if other types of CD122+ immune cells play any role in this experimental diabetes model.

5) It is not clear if CD122+ CD8 Treg cells were also elevated in pancreatic draining lymph nodes (PLN), as well as the pancreatic islets after infection. It is important to study the two sites, as PLN have close links with intestine, and pancreatic islets are the tissue where the inflammation and tissue damage occurs. Location matters.

6) The authors showed that, in the presence of APC, the suppressive function of CD8 Treg cells is evident; however, it is not clear what role the APCs play. It is important to investigate the role of APCs in the induction and function of these CD8 Tregs.

7) Human study: it is not clear if the healthy controls used in the study were age and sex matched as both factors contribute to the composition of gut microbiota. The authors stated the T1D patients were newly diabetic; however, the supplementary table showed that there was a wide range of disease onset time, from 0 month to 14 years plus 7 months. Moreover, 2 patients were toddlers, aged 2 and 3 years old. The diet of the young children must be different from the teens (~10/15 patients) and it is known that the diet is one of the major factors contributing to the composition of gut microbiota.

Reviewer #3 (Remarks to the Author):

In this study, Shimokawa and colleagues investigated the mechanism responsible for the protective effect of an intestinal nematode infection on experimental type I diabetes in mice. The authors show that *H. polygyrus*, a rodent model for chronic nematode infections, efficiently represses the development of streptozotocin-induced T1D.

While the prevention of experimental diabetes by Hp and other nematode infection models has been reported previously, the current study shows that the beneficial effect of the chosen infection model is linked to the expansion of CD8+ regulatory cells marked by the expression of the IL-2R β CD122. Depletion of the CD8+ regulatory subset reverts the protective effect of the nematode infection, whereas the adoptive transfer of the regulatory population isolated from nematode infected donors suppresses the development of the disease.

While the mechanism of protection against T1D mediated by CD8+ Treg is not clarified in the current study, the authors show that trehalose released by the parasite supports the outgrowth of CD8+ Treg. Importantly, feeding trehalose phenocopies the effect of the infection in supporting CD8+Treg and reducing the clinical disease parameters. Furthermore, NOD mice starting to display signs of beta-cell destruction can be rescued by trehalose feeding.

The authors show that the beneficial effects of the nematode infection depend on the microbiota. As both the infection and trehalose feeding lead to the increased abundance of Ruminococci, the authors conclude that sugar release by the parasite supports the outgrowth of these microbes, permitting the increase in CD8+ Treg controlling T1D. This is further supported by the reduced T1D development in mice fed high doses of a single Ruminococcus species and by increased CD8+Treg frequencies in cultures supplemented with products released by Ruminococci.

Hence, this study convincingly links the novel finding of trehalose release by an intestinal parasite to specific microbiota alterations, which in turn support the expansion and activity of CD8+ Treg cells. Of note, clinical data provided in the study show that diabetic children harbor fewer CD8+ Treg, relatively low abundances of Ruminococci and lower trehalose levels in blood than healthy control subjects. These findings suggest that deficits in trehalose, gut microbes and CD8+ Treg

may be involved in the development of human T1D and that trehalose supplementation or prebiotic-based strategies deserve to be considered as prophylactic and therapeutic applications.

While overall very well conducted and attractive to a broad readership, there are several points concerning the human as well as experimental parts that should be addressed by the authors before publication.

Major points:

1. Human data

1.1 A summary of the clinical characteristics of the healthy controls needs to be included. More specifically: Were the HC in the same age range as the TD1 patients? If the age range of the HC included older subjects and if the colonization/abundance of Ruminococcus increases with age, this might be related to the higher frequencies of CD8+ Treg detected in the HC cohort. This is even more important as Suppl. Fig. 7 shows that aged mice displaying increased frequencies of CD8+ Treg also harbor higher abundances of Ruminococcus.

1.2 Increased trehalose serum levels in HC cohort: This finding fits with the data on more CD8+Treg and elevated Ruminococcus abundances in HC. What could be the reason for the higher trehalose levels in healthy controls?

1.3 Fig 4A: Is this a common gating strategy for human CD8+Treg? Did the authors confirm that the CD122+CXCR3+ cells are 'real' CD8+ T cells by counterstaining e.g. TCR, CD45RA, CD62-L or CCR7? Human CD8+Treg are sometimes described as largely CD122 negative, please comment.

2. Local immune responses:

2.1 What's the authors suggestion for the mechanism of CD8+Treg mediated protection against beta-cell destruction? Did the authors investigate the responses in the murine pancreas in further detail? Were IFN-g or other markers of CD8+ T cell activity, such a granzyme B, surveyed in the pancreatic tissue via PCR and expressed in lower amounts in mice protected from disease development? Is it possible to co-stain CD8, CD122 and insulin in situ to see if CD8+ Treg locate around the islets in mice protected from experimental diabetes?

2.2 Supplementary Fig. 3: The figure legend states that pancreatic mononuclear cells were analyzed. The M&M flow cytometry section states that cells from pancreatic lymph nodes were used. Please clarify. How where the cells in Suppl. Fig. 3 stimulated to induce IFN-g production?

2.3 Was the frequency of CD8+Treg also surveyed in the pancreas/pancreatic lymph node tissue?

3. CD8+Treg origin/expansion

3.1 Have the authors assessed other organs next to the spleen for the frequencies of CD8+Tregs? Are they present in higher numbers in the small intestine or GALT of nematode infected mice?

3.2 Did the authors survey if the increased numbers of CD8+CD122+ Treg in Hp infected (or trehalose supplemented mice) originate from CD122- precursors or from expanding CD8+CD122+ T cells? This could be surveyed by adoptive transfer of CD8+CD122- cells, possibly combined with Ki-67 detection.

4. Generalization:

Several helminth infection models were shown to prevent or delay the onset of T1D. Did the authors check/inquire if also other nematode (e.g. extra-intestinal forms such as filaria reported to suppress T1D) release trehalose?

5. Effector cell suppression by CD8+ Treg in vitro/Figure 1:

5.1 Why do CD8+Treg not repress T cell responses in absence of APC? Do CD8+Treg depend on co-stimulation by APC which are activated by the effector T cells? Or do APC start to express inhibiting co-stimulatory molecules in response to TCR-mediated CD8+Treg activation, leading to poor T effector cell responses?

5.2 Does the suppressive effect of CD8+ Treg in vitro/in vivo relate to IL-2 consumption via IL-2Rb? Have the authors tried to add rIL-2 to see if the proliferation of CD4+/CD8+ effector cells can be rescued? What is the MFI of CD122 on CD8+ Treg from naïve/infected donor mice? Figure 1 D suggests that CD122 is expressed at higher levels by CD8+CD122+/low cells from infected donor mice.

Can CD8+Treg isolated from naïve donors be triggered to express higher levels of CD122, leading to similar efficiency in the suppressive assay as seen after culture with infection-derived CD8+Treg?

5.3 The authors state that the effect of IL-10 possibly made by CD8+Treg on T1D was limited. However, IL-10 may be responsible for the repressed T cell proliferation and IFN-g secretion in vitro. Did the authors try to block IL-10 in the suppression assays?

5.4 Fig 1: Please indicate how many times the assays depicted in Fig. 1J/K were performed. What is reflected by the SD? Inter-experimental variation or technical replicates?

6. Microbiota/Trehalose

The authors determined trehalose levels in the small intestine of mice, but the MB composition was surveyed based on fecal samples. Is Ruminococcus also present in higher numbers in the small intestine of infected mice? And is trehalose detectable in the large intestine?

7. Trehalose/CD8+Treg

In line 180, the authors state that 'trehalose does not appear to directly induce CD8+ Treg'. Did they rule out this possibility by e.g. in vitro work or by feeding trehalose to Abx-treated mice? HES-feeding for 7 days induces CD8+Treg, which is prevented by trehalase treatment of HES. Did the authors assess if HES (with and w/o trehalase treatment) can directly induce/support the differentiation of suppressive CD8+ Treg from CD8+ non-Treg cells in vitro?

Minor points:

1.

Fig 1j/k: The bar representing the proliferation/cytokine production by CD4/CD8 cells kept without CD8+ Treg should be changed to a different color/pattern for a better distinction from the responses in presence of CD8+ Treg from uninfected donors.

2.

The color coding needs to be reworked in some of the figures. Fig. 1e: H.p. = black according to legend, red in graph; Fig 2e: blue instead of green).

3.

Line 46-48: the last sentence is confusing, please rephrase.

4.

Please remove the typo in Fig. 3C (otheres).

Please readjust the color legend in 3E, it is partially hidden by the rest of the figure.

5.

The authors show that depletion of CD25+Treg does not prevent the protective effect of Hp infection. Did they control for co-depletion of CD25+ expressing CD8+ effector cells?

Are CD4+ Treg unaffected by a-CD122 treatment? Is it possible that both Treg populations

synergize in suppressing the onset of T1D?

6.

Fig. 1 K:

The y-axis scale in the IFN-g graphs should be changed to 2.5 as maximum.

The graphs depicting the IFN-g responses in CD4/CD8 cultures without APC, possibly coincidentally, look exactly the same. Please make sure that this is not an accidental duplication.

7.

Fig2C: Is it possible to express the trehalose amount in the SN of adult worms in relation to the amount of E/S proteins secreted by the worms?

8.

Fig 2J is not addressed in the text.

9.

Lines 133/134 suggest that Fig. 3D only includes members of the phylum Firmicutes. This contradicts with Akkermansia belonging to the class of Verrucomicrobiae and Desulfovibrio belonging to the class of delta-Proteobacteria.

10.

Line 148: this should probably read as '...closest relative...'

Reviewer #4 (Remarks to the Author):

Clarity is needed regarding the mass spectrometry metabolomics methods, data interpretation and results. The following is a list of elements that need to be addressed:

1. GC/MS platform for untargeted metabolomics reportedly identified 500 metabolites. Key advantage of the untargeted platform is the broad array of metabolites that are tested. However, little information is presented on the data output and integration. In the supplemental information, a table outlining all the metabolites identified with raw data should be presented.

2. What are the Qa/QC of the platform? How is the data normalized across the runs? What is the inter and intra assay variability? How many data points were missing? How is this missing data handled/ Was data imputed?

3. While the data reduction strategy using OPLS-DA is presented, it would be better to see individual metabolites (transformed data likely needs to be used as it is unlikely to be normally distributed) with Bonferroni for multiple comparison to identify differentially expressed metabolites. It is conceivable none meet the FDR threshold, in which case dimension reduction might be appropriate, however, it will be important for the authors to present the individual metabolite data first. This way, the data can be evaluated on the basis of the most important metabolic changes in addition to Trehalose which is explored in greater depth rest of the manuscript.

Reviewer #1 (Remarks to the Author):

General comment: The caveat of this paper is that streptozotocin-induced diabetes in rodents and autoimmune diabetes in NOD mice are not identical to human type 1 diabetes (T1D). Therefore I suggest that the title should be revised to read “CD8⁺ regulatory T cells play a critical role in prevention of streptozotocin –induced diabetes”.

We are really grateful for the reviewer’s careful reading and the constructive comments. As the reviewer suggested, we changed the title to “CD8⁺ regulatory T cells play a critical role in prevention of autoimmune-mediated diabetes” because we used NOD mice as well as STZ-treated mice.

Specific comments:

1.The methodology used seems appropriate.

We thank for the positive comments.

2. Do the authors have any information on the correlation between intestinal trehalose and trehalose in the peripheral circulation,

Thank the reviewers for this comment. In additional experiments, we measured serum trehalose concentrations in mice used, and the results are included as Fig. 2c. We can observe a close correlation between them and show the data below.

3. When looking at the supplementary table with data on the children with T1D analyzed, one can see that the disease duration is highly variable (from diagnosis to a disease duration of almost 15 years). Only two children have a disease duration of less than one year. It would have been preferable to include only children with newly diagnosed T1D

given that the abundance of CD8Treg cells is hardly stable over time.

We completely agree the reviewer's comment and tried to recruit such patients. However, unfortunately, the number of newly diagnosed children was limited, and we had analyzed these patients. We analyzed the correlation between duration from diagnosis and CD8Treg abundance. No relationship was observed, suggesting that T1D patients have lower number of CD8Treg indifferently of disease duration. The results are shown here only for the reviewer, and we think these results do not deny our proposal.

4. Ten of the 15 children with T1D have a CD8Treg count of $\geq 2\%$, which seems to be within the normal range based on Figure 4b.

If normal range is set as average ± 1 SD, CD8Treg count of 10 of the 15 patients looks within it. We thought it hard to define normal range and believe that comparing the average of each group is appropriate.

5. Only GAD antibodies were analyzed in the children with T1D. The cutoff for GAD antibody positivity should be given and IA-2 and ZnT8 antibodies should as well be analyzed in these children

The cutoff value for anti-GAD antibody is provided in Supplemental Table 3. We are very sorry that anti-ZnT8 antibody was not analyzed and anti-IA-2 antibody was analyzed only in patients negative for anti-GAD antibody, and the results were included in the Table.

6. From where is the thehalose in the human serum samples derived?

We think tolehalose in serum is derived from the dietary origin, and some foods are rich in trehalose. Furthermore, as shown in comment #1 serum trehalose closely correlates

with intestinal trehalose, suggesting trehalose is absorbed from the small intestines.

7. The proportional difference in the abundance of Ruminococcus in the gut microbiota between children with T1D and healthy children is conspicuously smaller than the difference in serum trehalose concentrations between the two groups. Why is that?

Base line of the relative abundance of Ruminococcus and serum trehalose is 11.3 % and 0.74 mg/ml healthy children, respectively. Those are dropped to 1.2 % and 0.07 mg/ml in T1D children. The proportional difference is not so different. We think those graphs seem to be visually different.

Reviewer #2 (Remarks to the Author):

The finding that helminthic infection can protect from type 1 diabetes in mouse models is not novel, however, the novelties of this manuscript are as follows: 1) the authors identified that CD122⁺ CD8⁺ T cells induced by a mouse nematode (*Heligmosomoides polygyrus*) infection contribute to the protection from diabetes; 2) the authors identified that trehalose, possibly produced by gut commensal bacteria *Ruminococcus*, contributes to CD8 Treg induction and diabetes protection and 3) the authors also found that children with T1D have lower levels of CD122⁺ CD8 Treg and lower abundance of *Ruminococcus* in the fecal samples. However, this reviewer has the following concerns and comments:

We are really grateful for the reviewer's careful reading and the constructive comments.

1) This is not an ideal model: the author used a chemical, STZ, to generate an induced diabetes model, which is not an autoimmune diabetes model. The best mouse model of type 1 diabetes is the non-obese diabetic (NOD) mouse model, and although the STZ model system has been used by many investigators, the cause of human T1D is not by a beta cell toxic chemical agent. Thus additional cautions apply in the translation of the finding to humans.

Thank you for the comment. As the reviewer pointed out, low-dose multiple STZ model is generally accepted as an autoimmune-mediated diabetes, and we also believe STZ model could be appropriate. In contrast to high-dose injection, low-dose is much less toxic and slightly destroys beta cells, which may allow to evoke immune responses to "hidden" beta cell antigens. We also analyzed NOD mice in detail as the reviewers suggested (see below), and similar results were obtained.

2) In one experiment, the authors used diabetic NOD mice and found that trehalose feeding could reverse the hyperglycemia for up to a month in about 50% of diabetic NOD mice. It is not clear if the mice that became normoglycemic expressed more CD122⁺ CD8 Tregs; and it is also not clear if there were changes in the composition of gut microbiota of these mice.

Following the reviewer's comment, we analyzed CD8Treg and *Ruminococcus* in NOD

mice. These results are included as Fig. 2m, and Supplemental Figure 9, respectively. We really thank the reviewer, and these results support our proposal. However, trehalose treatment increased Ruminococcus and CD8Treg even in mice refractory to the treatment, suggesting that severe CD8Tregs could prevent diabetes before beta cells are totally destroyed but not reverse after beta cell loss.

3) STZ needs to be dissolved in a citrate acid buffer, pH 4.0, and studies have suggested that pH, especially low pH can alter the composition of gut microbiota. The authors should use citrate buffer, not PBS (pH ~7.4), as the control, especially as the authors studied the gut microbiota in the mice.

To avoid such influence by acidic agents even it is injected intraperitoneally, we used STZ freshly dissolved in PBS. Thus, we think that PBS is an appropriate control.

4) CD122 is expressed on other types of immune cells including NK cells; it is not clear what the effect of anti-CD122 antibody depletion in vivo was on other types of immune cells, in addition to the CD122+ CD8 T cells. It is also not clear if other types of CD122+ immune cells play any role in this experimental diabetes model.

As shown in the original Fig. 1d, anti-CD122 depletes not only CD8Treg but also other cells expressing CD122. To exclude a possibility that CD122-expressing cells other than CD8Tregs play suppressive roles in T1D development, CD8Tregs purified from Hp-infected mice were transferred to recipient mice. Our results indicate transfer of CD8Treg confers protection from T1D onset.

5) It is not clear if CD122+ CD8 Treg cells were also elevated in pancreatic draining lymph nodes (PLN), as well as the pancreatic islets after infection. It is important to study the two sites, as PLN have close links with intestine, and pancreatic islets are the tissue where the inflammation and tissue damage occurs. Location matters.

We analyzed CD8Tregs in PLN and the pancreatic tissues and found that CD8Tregs were increased in PLN and the results are included as a new Fig 1d, e. However, the increase in CD8Treg was less obvious and thus, we could not conclude that. As shown in

Supplementary Figure 3, infection with Hp decreased pathogenic T cells in pancreas in a CD8Treg-dependent manner. CD8Tregs are not necessarily increased in situ and it is possible that CD8Treg suppresses priming of pathogenic T cells in the draining lymphoid organs. We will analyze these issues in detail to understand where and how CD8Tregs suppress pathogenic T cells and their suppressive mechanisms in the future.

6) The authors showed that, in the presence of APC, the suppressive function of CD8 Treg cells is evident; however, it is not clear what role the APCs play. It is important to investigate the role of APCs in the induction and function of these CD8 Tregs.

Similar to CD4Tregs, CD8Tregs require APCs for their suppressing activities. However, roles of APCs are not evaluated in this manuscript, and they will be analyzed in the next manuscript.

7) Human study: it is not clear if the healthy controls used in the study were age and sex matched as both factors contribute to the composition of gut microbiota. The authors stated the T1D patients were newly diabetic; however, the supplementary table showed that there was a wide range of disease onset time, from 0 month to 14 years plus 7 months. Moreover, 2 patients were toddlers, aged 2 and 3 years old. The diet of the young children must be different from the teens (~10/15 patients) and it is known that the diet is one of the major factors contributing to the composition of gut microbiota.

We are very sorry that we did not describe healthy volunteers' information in detail and included them in a new Supplemental Table 3. Although the abundance of Ruminococcus and the amount of CD8Tregs seems to be affected age in healthy donors, those are still low indifferently of age and duration from the onset in T1D patients. To minimize the effects of age and sex, we recruited age- and sex-matched healthy volunteers. The relationship between CD8Tregs/Ruminococcus and age/duration is shown for the reviewers.

Reviewer #3 (Remarks to the Author):

While overall very well conducted and attractive to a broad readership, there are several points concerning the human as well as experimental parts that should be addressed by the authors before publication.

We are really grateful for the reviewer's careful reading and the constructive comments.

Major points:

1. Human data

1.1 A summary of the clinical characteristics of the healthy controls needs to be included. More specifically: Were the HC in the same age range as the TD1 patients? If the age range of the HC included older subjects and if the colonization/abundance of Ruminococcus increases with age, this might be related to the higher frequencies of CD8+ Treg detected in the HC cohort. This is even more important as Suppl. Fig. 7 shows that aged mice displaying increased frequencies of CD8+ Treg also harbor higher abundances of Ruminococcus.

We are very sorry that we did not describe healthy volunteers' information in detail and included them in a new supplemental table 1. Although the abundance of Ruminococcus seems to be affected age in healthy donors as, that is still low indifferently of age in T1D patients. To minimize the effects of age and sex, we recruited age- and sex-matched healthy volunteers. The relationship between CD8Tregs/Ruminococcus and age/duration is shown for the reviewers.

1.2 Increased trehalose serum levels in HC cohort: This finding fits with the data on more CD8+Treg and elevated Ruminococcus abundances in HC. What could be the reason for the higher trehalose levels in healthy controls?

The question is very interesting. Trehalose may be absorbed from diet intake and some are digested by trehalase. But, unfortunately, we do not know the reason why HC contains high or T1D patients contains low trehalose. Food intake, absorption, and trehalase activities may cause the differences.

1.3 Fig 4A: Is this a common gating strategy for human CD8⁺Treg? Did the authors confirm that the CD122⁺CXCR3⁺ cells are 'real' CD8⁺ T cells by counterstaining e.g. TCR, CD45RA, CD62-L or CCR7? Human CD8⁺Treg are sometimes described as largely CD122 negative, please comment.

Thank the reviewer's comment. We did not confirm by counterstaining. Indeed, there are controversial reports for CD8Treg's phenotype, some are CD122⁻ and others are CD122⁺. We could not definitely mention that CXCR3⁺CD8^{dull}CD122⁺ cells are CD8Tregs because we did not perform suppression assays using those cells. However, according to the publication below we think those cells are CD8Treg.

Li S. et al. A naturally occurring CD8⁺CD122⁺ T-cell subset as a memory-like Treg family. *Cellular & Molecular Immunology* 11: 326-331, 2014.

2. Local immune responses:

2.1 What's the authors suggestion for the mechanism of CD8⁺Treg mediated protection against beta-cell destruction? Did the authors investigate the responses in the murine pancreas in further detail? Were IFN-g or other markers of CD8⁺ T cell activity, such a granzyme B, surveyed in the pancreatic tissue via PCR and expressed in lower amounts in mice protected from disease development?

It is an important question. We do not know the suppressive mechanisms. As shown in the original Fig. S3, CD8Tregs may attenuate IFN-g response of pathogenic CD8 T cells. However, other markers of have not been evaluated. Please also see our responses to #2.3 below.

Is it possible to co-stain CD8, CD122 and insulin in situ to see if CD8⁺ Treg locate around the islets in mice protected from experimental diabetes?

It would be very interesting to see suppression in situ. However, we did not perform the experiments because we do not know the exact location where immune suppression occurs, for instance pancreatic islet in situ or lymphoid tissues as priming site for pathogenic T cells.

2.2 Supplementary Fig. 3: The figure legend states that pancreatic mononuclear cells were analyzed. The M&M flow cytometry section states that cells from pancreatic lymph nodes were used. Please clarify. How were the cells in Suppl. Fig. 3 stimulated to induce IFN- γ production?

We are sorry for confusion. Pancreatic mononuclear cells are correct and we added this in M & M. We added the stimulation, plate-bound anti-CD3.

2.3 Was the frequency of CD8⁺Treg also surveyed in the pancreas/pancreatic lymph node tissue?

Yes. We analyzed CD8Tregs in PLN and the pancreatic tissues and found that CD8Tregs were increased in PLN and the results are included as a new Fig 1d, e. However, the increase in CD8Treg was less obvious and thus, we could not conclude that. As shown in Supplementary Figure 3, infection with Hp decreased pathogenic T cells in pancreas in a CD8Treg-dependent manner. CD8Tregs are not necessarily increased in situ and it is possible that CD8Treg suppresses priming of pathogenic T cells in the draining lymphoid organs. We will analyze these issues in detail to understand where and how CD8Tregs suppress pathogenic T cells and their suppressive mechanisms in the future.

3. CD8⁺Treg origin/expansion

3.1 Have the authors assessed other organs next to the spleen for the frequencies of CD8⁺Tregs? Are they present in higher numbers in the small intestine or GALT of nematode infected mice?

We analyzed mesenteric LN and found Hp infection increases CD8Tregs.

3.2 Did the authors survey if the increased numbers of CD8+CD122+ Treg in Hp infected (or trehalose supplemented mice) originate from CD122- precursors or from expanding CD8+CD122+ T cells? This could be surveyed by adoptive transfer of CD8+CD122- cells, possibly combined with Ki-67 detection.

It is also important question. Our transfer experiments revealed that CD8+CD122+ cells protect the recipient mice, suggesting developed CD8Treg may divide and function in the recipients. This report focuses on protective roles and induction of CD8Treg rather than differentiation and/or expansion, and this will be investigated in the future.

4. Generalization:

Several helminth infection models were shown to prevent or delay the onset of T1D. Did the authors check/inquire if also other nematode (e.g. extra-intestinal forms such as filaria reported to suppress T1D) release trehalose?

We have not analyzed other helminthic infections. A free-living nematode, *C. elegans* is known to produce trehalose probably in order to adapt environmental conditions. As parasitic nematodes experience lifecycle in external environment, they may possess the ability to produce cytoprotective trehalose.

5. Effector cell suppression by CD8+ Treg in vitro/Figure 1:

5.1 Why do CD8+Treg not repress T cell responses in absence of APC? Do CD8+Treg depend on co-stimulation by APC which are activated by the effector T cells? Or do APC start to express inhibiting co-stimulatory molecules in response to TCR-mediated CD8+Treg activation, leading to poor T effector cell responses?

CD8Tregs are known to mediate immune suppression through various cell contact-dependent and –independent mechanisms. CD8Tregs induced in Hp infection may function via cell contact with APC. However, the precise mechanisms are not evaluated in this manuscript and will be investigated in the future.

5.2 Does the suppressive effect of CD8+ Treg in vitro/in vivo relate to IL-2 consumption

via IL-2Rb? Have the authors tried to add rIL-2 to see if the proliferation of CD4⁺/CD8⁺ effector cells can be rescued? What is the MFI of CD122 on CD8⁺ Treg from naïve/infected donor mice? Figure 1 D suggests that CD122 is expressed at higher levels by CD8⁺CD122⁺/low cells from infected donor mice.

Can CD8⁺Treg isolated from naïve donors be triggered to express higher levels of CD122, leading to similar efficiency in the suppressive assay as seen after culture with infection-derived CD8⁺Treg?

According to the comment we analyzed expression of CD122 in CD8Treg before and after infection with Hp. As the reviewer pointed out, expression of CD122 is enhanced in Hp-infected mice. Our results suggested that bacterial products activate CD8Treg in vitro, and we plan to analyze bacterial metabolite to induce CD122 expression. Thus, bacteria rather than infection-derived (activated) CD8Treg are likely to induce it.

Although it is important to know the involvement of IL-2 in CD8Treg function, we are very sorry that we did not address these issues the reviewer raised.

5.3 The authors state that the effect of IL-10 possibly made by CD8⁺Treg on T1D was limited. However, IL-10 may be responsible for the repressed T cell proliferation and IFN-g secretion in vitro. Did the authors try to block IL-10 in the suppression assays?

We are sorry that we did not use anti-IL-10 in in vitro experiments because it did not reverse suppressive effects in vivo.

5.4 Fig 1: Please indicate how many times the assays depicted in Fig. 1J/K were performed. What is reflected by the SD? Inter-experimental variation or technical replicates?

The assays were repeated three times. And SD is calculated from triple or quadruple cultured, which is clearly described in the legend.

6. Microbiota/Trehalose

The authors determined trehalose levels in the small intestine of mice, but the MB composition was surveyed based on fecal samples. Is Ruminococcus also present in

higher numbers in the small intestine of infected mice? And is trehalose detectable in the large intestine?

We analyzed the composition of microbiota in the small intestines, and the results were included as Fig. 3c. It shows Ruminococcus is increased in the small intestines at greater degree than in feces. Trehalose is hard to detect in the large intestines because it is absorbed or digested in the small intestines. Thus, Ruminococcus might be increased mainly in the small intestines.

7. Trehalose/CD8⁺Treg

In line 180, the authors state that ‘trehalose does not appear to directly induce CD8⁺ Treg’. Did they rule out this possibility by e.g. in vitro work or by feeding trehalose to Abx-treated mice?

HES-feeding for 7 days induces CD8⁺Treg, which is prevented by trehalase treatment of HES. Did the authors assess if HES (with and w/o trehalase treatment) can directly induce/support the differentiation of suppressive CD8⁺ Treg from CD8⁺ non-Treg cells in vitro?

We did not use trehalose and HES to induce CD8Tregs in vitro, because CD8Tregs were not induced in the absence of intestinal microbiota even in the presence of trehalose or Hp.

Minor points:

1.

Fig 1j/k: The bar representing the proliferation/cytokine production by CD4/CD8 cells kept without CD8⁺ Treg should be changed to a different color/pattern for a better distinction from the responses in presence of CD8⁺ Treg from uninfected donors.

Following the suggestion, we changed the bar graphs.

2.

The color coding needs to be reworked in some of the figures. Fig. 1e: H.p. = black according to legend, red in graph; Fig 2e: blue instead of green).

We are sorry for the mistakes and corrected them.

3.

Line 46-48: the last sentence is confusing, please rephrase.

We corrected in the revision, thank you.

4.

Please remove the typo in Fig. 3C (otheres).

Please readjust the color legend in 3E, it is partially hidden by the rest of the figure.

We corrected in the revision, thanks again.

5.

The authors show that depletion of CD25⁺Treg does not prevent the protective effect of Hp infection. Did they control for co-depletion of CD25⁺ expressing CD8⁺ effector cells? Are CD4⁺ Treg unaffected by a-CD122 treatment? Is it possible that both Treg populations synergize in suppressing the onset of T1D?

These possibilities are not experimentally excluded. However, protective roles of CD8Tregs were confirmed in the transfer experiments, and we think CD8Tregs alone are responsible for the suppressive effects.

6.

Fig. 1 K:

The y-axis scale in the IFN-g graphs should be changed to 2.5 as maximum.

The graphs depicting the IFN-g responses in CD4/CD8 cultures without APC, possibly coincidentally, look exactly the same. Please make sure that this is not an accidental duplication.

We corrected the y-axis scale in the revision. We are sorry for mistake and changed to the right ones.

7.

Fig2C: Is it possible to express the trehalose amount in the SN of adult worms in relation to the amount of E/S proteins secreted by the worms?

We are sorry that we did not measure protein concentrations in HES.

8.

Fig 2J is not addressed in the text.

We are afraid that we addressed Fig. 2j on line 120 to 122 in the original version.

9.

Lines 133/134 suggest that Fig. 3D only includes members of the phylum Firmicutes. This contradicts with Akkermansia belonging to the class of Verrucomicrobiae and Desulfovibrio belonging to the class of delta-Proteobacteria.

We removed the original Fig 3c showing microbiota composition in the fecal samples at the phylum level and instead include data of that in the small intestines at the genus level in response to the reviewer's major comment #6.

10.

Line 148: this should probably read as '...closest relative...'.
We corrected properly, thanks.

Reviewer #4 (Remarks to the Author):

Clarity is needed regarding the mass spectrometry metabolomics methods, data interpretation and results. The following is a list of elements that need to be addressed:

We are really grateful for the reviewer's careful reading and the constructive comments.

1. GC/MS platform for untargeted metabolomics reportedly identified 500 metabolites. Key advantage of the untargeted platform is the broad array of metabolites that are tested. However, little information is presented on the data output and integration. In the supplemental information, a table outlining all the metabolites identified with raw data should be presented.

We are sorry for not providing sufficient information. Our GC/MS platform is capable of detecting more than 500 metabolites, and we finally identified 48 and 33 metabolites in contents of the small intestines and HES, respectively using MRM analysis. We added tables outlining all the metabolites as new Supplementary Table 1 and 2. And we modified in the text.

2. What are the Qa/QC of the platform? How is the data normalized across the runs? What is the inter and intra assay variability? How many data points were missing? How is this missing data handled/ Was data imputed?

2-IPM was added to all the samples. QC sample was made by mixing all the target samples. Four QC measurements were done in the case of both intestinal contents and HES. %CV of 2-IPM area was examined to assure the quality of the measurements. %CV of 2-IPM area among the 4 QC measurements were 8.9% and 19% in the small intestines and HES, respectively. %CV of each metabolite were also presented in the Supplementary Table 1 and 2. The variation of retention time was also presented to confirm the identification of the metabolites.

In the case of the analysis of intestinal contents, we normalized the data using the peak area of 2-IPM and the weight of intestinal contents. In the case of the analysis of HES, we normalized the data using the peak area of 2-IPM.

The information about the Qa/QC above was described in the supplementary tables.

Intra assay variability was evaluated by the measurement of n-alkane mixture. The sensitivity and the set value of RT were adjusted before the measurement.

We watched every chromatographic peak and revised manually. In the Supplementary Table 1 and 2, there is no data missing data points.

3. While the data reduction strategy using OPLS-DA is presented, it would be better to see individual metabolites (transformed data likely needs to be used as it is unlikely to be normally distributed) with Bonferroni for multiple comparison to identify differentially expressed metabolites. It is conceivable none meet the FDR threshold, in which case dimension reduction might be appropriate, however, it will be important for the authors to present the individual metabolite data first. This way, the data can be evaluated on the basis of the most important metabolic changes in addition to Trehalose which is explored in greater depth rest of the manuscript.

According to the reviewer's suggestion, Bonferroni and FDR (Benjamini-Hochberg) method was adapted to the data analysis, and the data are included as new Fig. 2a, and d. As shown in the Supplementary Tables, trehalose was significantly high both in the intestinal content of the infected mice and HES using Bonferroni method.

Reviewers' comments:

Reviewer #1 (Remarks to the Author):

The authors have responded appropriately to most of the comments by the reviewers. In their response to my third comment they tell that "no relationship was observed between the disease duration and the abundance of CD8Tregs". They include in their response two figures showing the correlation between the abundance of CD8Tregs and disease duration to the right and between the abundance of Ruminococcus and disease duration to the left. They tell that the R2 for the first correlation is - (?) 0.667 and for the second correlation 0.744. This indicates that the correlation coefficient is -0.817 for the first correlation and 0.863 for the second correlation. These correlation coefficients must be statistically significant.

The language of the manuscript should be checked by an experienced person with English as his/her native language.

Reviewer #2 (Remarks to the Author):

The authors have done some additional experiments including the experiments using NOD mice. Overall the results from the additional experiments support their initial finding. This reviewer has some comments to the revised manuscript.

Major

1) It is not scientifically rigorous to consider STZ-induced diabetes to be autoimmune-mediated diabetes or type 1 diabetes, even when using multiple "low" dose STZ injection. STZ when given as low dose, multiple injections, induces diabetes even in immune-deficient mice (Gerling et al., Diabetes 43:433-40, 1994; Reddy, et al., Autoimmunity 20:83-92, 1995; Chaudhry, et al., Laboratory Animals 47:257, 2013). Moreover, STZ has also been used in rodent models for obesity/type 2 diabetes studies (Sugano et al., Nutrition, Metabolism & Cardiovascular Diseases 16:477-484, 2006; Zhang, et al, Exp Diab Res. Doi:10.1155/2008/704045; Srinivasan et al., Pharmacol Res 52:313-20, 2005; Barriere et al., Sci Rep 8:424, 2018). Thus, we need to be more cautious about usage of the term, as it may mislead both scientific and lay public readers. STZ is a toxic agent that specifically targets insulin-producing beta cells, leading to hyperglycemia, i.e., diabetes.

2) It appears that the authors used the same anti-CD122 antibody (TM β -1) for in vivo depletion and in vitro detection of depletion efficiency (Fig.1f). Since the half life of rat IgG is 5-8 days, it is highly likely that the antibody is still bind on its target, IL-2R β , using the same antibody that recognizes the same epitope of the target antigen potentially over estimate the depletion efficiency. Better scientific approach is to use a different mAb that also recognizes IL-2R β but different epitope from TM β -1.

Minor

1) Given the comments from Reviewer #1 and the comments above, this reviewer suggests that the authors replace STZ-induced T1D to STZ-induced diabetes, throughout the manuscript.

2) Fig.2 m, the authors indicated that NOD mice were used for the experiment. NOD mice ARE wild type, the same as C57BL/6 mice, but a different strain with different genetic background from C57BL/6. Hence, it is confusing what does WT (wild type) mean on the X-axis.

3) The gender of the mice used in the study is not clear. This reviewer assumed that male C57BL/6 mice were used for STZ experiments, as male mice are more susceptible to STZ-induced diabetes, whereas female NOD mice were used for the NO experiments, as female NOD mice are more prone to develop spontaneous autoimmune diabetes. The authors should clarify the mouse gender used in the study.

4) Fig.3a, the authors stated the percentage of CD8+CD122+ cells in gated lymphoid cells. Please indicate what maker was used in gating lymphoid cells.

Reviewer #3 (Remarks to the Author):

I have no further objections, as the authors provide additional information on the human cohort, murine CD8+ Treg in the pancreatic lymph node and microbiota alterations in the small intestine of nematode infected mice.

Reviewer #4 (Remarks to the Author):

The authors have addressed all my comments and the work should now be acceptable for publication

Reviewer #1

The authors have responded appropriately to most of the comments by the reviewers. In their response to my third comment they tell that "no relationship was observed between the disease duration and the abundance of CD8Tregs". They include in their response two figures showing the correlation between the abundance of CD8Tregs and disease duration to the right and between the abundance of *Ruminococcus* and disease duration to the left. They tell that the R2 for the first correlation is - (?) 0.667 and for the second correlation 0.744. This indicates that the correlation coefficient is -0.817 for the first correlation and 0,863 for the second correlation. These correlation coefficients must be statistically significant.

We are very grateful for the reviewer's careful reading

First of all, we are very sorry that we mislabeled R to R2 value in the Figures we showed in the responses. Thus, the correct R2 value for the correlation between the abundance of CD8Tregs and duration from onset is 0.44 and that between the abundance of *Ruminococcus* and duration from onset is 0.55 that still suggests the significant relationship. However, we did not further examine the relationship and do not comment on this issue in this manuscript.

The language of the manuscript should be checked by an experienced person with English as his/her native language.

We omitted proofreading of the added parts of the revision and have the revision proofread. The edited parts are written in red.

Reviewer #2

The authors have done some additional experiments including the experiments using NOD mice. Overall the results from the additional experiments support their initial finding. This reviewer has some comments to the revised manuscript.

We are very grateful for the reviewer's careful reading and positive considerations.

Major

1) It is not scientifically rigorous to consider STZ-induced diabetes to be autoimmune-mediated diabetes or type 1 diabetes, even when using multiple low-dose STZ injection. STZ when given as low dose, multiple injections, induces diabetes even in immune-deficient mice (Gerling et al., Diabetes 43:433-40, 1994; Reddy, et al., Autoimmunity 20:83-92, 1995; Chaudhry, et al., Laboratory Animals 47:257, 2013). Moreover, STZ has also been used in rodent models for obesity/type 2 diabetes studies (Sugano et al., Nutrition, Metabolism & Cardiovascular Diseases 16:477-484, 2006; Zhang, et al, Exp Diab Res. Doi:10.1155/2008/704045; Srinivasan et al., Pharmacol Res 52:313-20, 2005; Barriere et al., Sci Rep 8:424, 2018). Thus, we need to be more cautious about usage of the term, as it may mislead both scientific and lay public readers. STZ is a toxic agent that specifically targets insulin-producing beta cells, leading to hyperglycemia, i.e., diabetes.

As the reviewer pointed out, multiple low-dose STZ were used to induce type 2 diabetes in conjunction with high-fat diet-induced obesity. However we still believe multiple low-dose STZ alone is for immune-mediated insulin-dependent diabetes as used in several publications.

2) It appears that the authors used the same anti-CD122 antibody (TM β -1) for in vivo depletion and in vitro detection of depletion efficiency (Fig.1f). Since the half life of rat IgG is 5-8 days, it is highly likely that the antibody is still bind on its target, IL-2R β , using the same antibody that recognizes the same epitope of the target antigen potentially over estimate the depletion efficiency. Better scientific approach is to use a different mAb that also recognizes IL-2R β ; but different epitope from TM β -1.

As pointed out, CD8Tregs were detected using TM β -1, the same antibody used for depletion. We reevaluate using a clone 5H4 and found that CD8Tregs were obviously depleted but a part of them still existed (attached figures). We think that the depleted cells contributed to suppression of STZ-induced diabetes even the depletion effects were partial. To further confirm the preventive roles of CD8Tregs, transfer experiments were performed.

Minor

1) Given the comments from Reviewer #1 and the comments above, this reviewer suggests that the authors replace STZ-induced T1D to STZ-induced diabetes, throughout the manuscript.

We reworded STZ-induced T1D to STZ-induced diabetes as the reviewer suggested.

2) Fig.2 m, the authors indicated that NOD mice were used for the experiment. NOD mice ARE wild type, the same as C57BL/6 mice, but a different strain with different genetic background from C57BL/6. Hence, it is confusing what does WT (wild type) mean on the X-axis.

We are sorry for confusion. We relabeled as B6 mice.

3) The gender of the mice used in the study is not clear. This reviewer assumed that male C57BL/6 mice were used for STZ experiments, as male mice are more susceptible to STZ-induced diabetes, whereas female NOD mice were used for the NO experiments, as female NOD mice are more prone to develop spontaneous autoimmune diabetes. The authors should clarify the mouse gender used in the study.

We already clearly described the mouse gender used in this study in Online Methods section. For more clarification, we added them in the legends, line 300 and 344.

4) Fig.3a, the authors stated the percentage of CD8+CD122+ cells in gated lymphoid

cells. Please indicate what marker was used in gating lymphoid cells.

Lymphoid cells were gated using FSC (forward scatter: cells size) and SSC (side scatter: granularity), which is indicated in the legend, line 308.

Reviewer #3

I have no further objections, as the authors provide additional information on the human cohort, murine CD8⁺ Treg in the pancreatic lymph node and microbiota alterations in the small intestine of nematode infected mice.

Reviewer #4 (Remarks to the Author):

The authors have addressed all my comments and the work should now be acceptable for publication

We are very grateful for the reviewers' careful reading and positive considerations.

REVIEWERS' COMMENTS:

Reviewer #2 (Remarks to the Author):

- 1) This reviewer suggests the authors to change the title of their article to "CD8+ regulatory T cells play a critical role in prevention of low dose of STZ induced diabetes"
- 2) Based on the new information provided by the authors, the CD8 depletion was not complete. Although the outcome did not change, this reviewer suggests that the authors should mention this in the text.

Reviewer #2 (Remarks to the Author):

1) This reviewer suggests the authors to change the title of their article to "CD8+ regulatory T cells play a critical role in prevention of low dose of STZ induced diabetes"

We thank the reviewer for careful reading.

We analyzed NOD mice and patients with T1D as well as STZ-treated C57BL/6 mice. Thus, because our findings are not limited to STZ-induced diabetes, we think that the title the reviewer suggested seems not appropriate and that the current one is better.

2) Based on the new information provided by the authors, the CD8 depletion was not complete. Although the outcome did not change, this reviewer suggests that the authors should mention this in the text.

We could not find the importance of mentioning the incompleteness of cell depletion that has not affected the outcome. To further confirm the critical roles of CD8Tregs, adoptive transfer experiments were performed. Thus, we are sorry that we do not mention this information.